# Origin of the multi-phasic quenching dynamics in the BLUF domains across the species

Yalin Zhou[1,3], Siwei Tang[1,3], Zijing Chen[1], Zhongneng Zhou[1], Jiulong Huang[1], Xiu-Wen Kang[1], Shuhua Zou[1], Bingyao Wang[1], Tianyi Zhang [1], Bei Ding [1] ✉ & Dongping Zhong [1,2] ✉

Blue light using flavin (BLUF) photoreceptors respond to light via one of nature's smallest photo-switching domains. Upon photo-activation, the flavin cofactor in the BLUF domain exhibits multi-phasic dynamics, quenched by a proton-coupled electron transfer reaction involving the conserved Tyr and Gln. The dynamic behavior varies drastically across different species, the origin of which remains controversial. Here, we incorporate site-specific fluorinated Trp into three BLUF proteins, *i.e.*, AppA, *Oa*PAC and *Sy*PixD, and characterize the percentages for the $W_{out}$, $W_{in}NH_{in}$ and $W_{in}NH_{out}$ conformations using $^{19}F$ nuclear magnetic resonance spectroscopy. Using femtosecond spectroscopy, we identify that one key $W_{in}NH_{in}$ conformation can introduce a branching one-step proton transfer in AppA and a two-step proton transfer in *Oa*PAC and *Sy*PixD. Correlating the flavin quenching dynamics with the active-site structural heterogeneity, we conclude that the quenching rate is determined by the percentage of $W_{in}NH_{in}$, which encodes a Tyr-Gln configuration that is not conducive to proton transfer.

Exquisite protein machines achieve a variety of functionalities by dynamically sampling various conformations that are energetically accessible[1,2]. Hence, structural heterogeneities are ubiquitously found in photoreceptors and enzymes, such as in green fluorescence proteins (GFP)[3,4], phytochromes[5,6], microbial rhodopsins[7], dihydrofolate reductases (DHFR)[8], blue light using flavin (BLUF) domains[9–11], and liver alcohol dehydrogenases (LADH)[8]. Quantifying structural heterogeneities and elucidating how they affect the chemical reaction dynamics in photoreceptors and enzymes are key to the rational de novo design for protein machines beyond those presented by nature[1,12,13].

Among the above proteins, the BLUF domain consists of a central flavin mononucleotide (FMN)-Gln-Tyr triad, which undergoes a photo-induced bidirectional proton-coupled electron transfer (PCET) reaction and then locks the protein in a meta-stable signaling state with the hydrogen bond (H-bond) network around FMN rearranged[14–16]. The quenching of the photo-excited FMN* during the photoactivation exhibits diverse and multi-exponential decay dynamics in BLUF domains of different species, the structural origin of which remains controversial[15–17]. For *Sy*PixD, the majority of the FMN* quenches rapidly in a few picoseconds, with large amounts of the diradical state intermediate FMNH˙/YO˙ detected in the photocycle, using both transient UV/Vis and mid-infrared spectroscopy[18–20]. On the other hand, no discernable intermediates have been captured in AppA so far, which features a nanosecond slow FMN* decay[17,21–23]. More recently, our group and the Lukacs, Tonge, and Meech groups have discovered that another BLUF domain, *Oa*PAC, exhibits a moderate FMN* decay (~344 ps), with a smaller amount of FMNH˙/YO˙ compared to *Sy*PixD in the photocycle[14,16,24]. The BLUF domain is widely present in proteins of many microorganisms, and their downstream functions vary, but

[1]Center for Ultrafast Science and Technology, School of Chemistry and Chemical Engineering, Shanghai Jiao Tong University, Shanghai 200240, China. [2]Department of Physics, Department of Chemistry and Biochemistry, and Programs of Biophysics, Chemical Physics, and Biochemistry, The Ohio State University, Columbus, OH 43210, USA. [3]These authors contributed equally: Yalin Zhou, Siwei Tang. ✉e-mail: bei.ding@sjtu.edu.cn; zhong.28@osu.edu

despite potential differences in downstream functions, the photo-switching functionality of the BLUF domain itself is remarkably robust in all known species. Hence, those three BLUF domains serve as an excellent comparison in interrogating the structural basis of the complex quenching dynamics. This understanding of diverse FMN* relaxation dynamics is crucial for our ultimate goal of unraveling the photoisomerization properties of BLUF domains, which can lead to the development of tailor-made optogenetic tools.

Recent free-energy simulation work performed by the Hammes-Schiffer group correlated the quenching dynamics to the active site Tyr-Gln geometry and the configurations of a nearby semi-conserved Trp[17,25,26]. As shown in Fig. 1, their simulation shows that when the nearby Trp is present as in a $W_{out}$ or $W_{in}NH_{out}$ conformation, Tyr OH is facing towards the Gln O$\varepsilon$1, and Gln N$\varepsilon$2H is well aligned with FMN N5, forming an H-bond thread conducive to proton relay, providing a plausible explanation for the rapid FMN* decay in SyPixD. On the other hand, when the nearby Trp adopts a $W_{in}NH_{in}$ configuration, the central Tyr and Gln are prone to orient in a no proton relay geometry, presumably leading to a slow FMN* decay, as observed in AppA. However, experimental validation of the above notion is highly challenging because the active-site H-bond network cannot be well-characterized using conventional X-ray crystallography. With limited resolution (typically 2–3 Å), X-ray crystallography cannot distinguish the two Gln orientations in Fig. 1a–c due to the similar size for the electron densities of Gln O$\varepsilon$1 and N$\varepsilon$2 atoms[17,25,27–31]. Besides, although structural heterogeneity in BLUF domains has been predicted according to the protein energetics calculations, X-ray structures are subject to crystal lattice confinement and may not accurately reflect the conformation distributions in solution[32]. Despite fluorescence measurements performed to interrogate the Trp conformations in aqueous solutions[33–35], the relation between Trp conformation and the H bond network in the active site remains largely unknown.

In this work, we substituted the nearby Trp in BLUF domains into various fluorinated Trp using the auxotrophic strain method and then utilized $^{19}$F nuclear magnetic resonance spectroscopy (NMR) to quantify the static structural heterogeneities in the solution. The molecular nature of the Trp conformations is identified by combining the techniques of $^{19}$F NMR and ultrafast spectroscopy with quantified proportions for $W_{in}NH_{in}$, $W_{in}NH_{out}$, and $W_{out}$ conformations in each species. Our methodology combining ultrafast transient absorption (TA) spectroscopy and $^{19}$F NMR herein can be used to address the

ubiquitous "heterogeneity problem" in photoreceptors and other protein machines[2,36].

## Results

### Solution NMR with site-specific 5-fluorotryptophan probes

Previous free-energy calculations[17,25] on the dark-state AppA and SyPixD BLUF domain have disclosed a relation between the nearby Trp conformations and the central Gln rotamers, indicating that the nearby Trp conformer could potentially serve as a readout of Tyr-Gln H-bond patterns. Instead of using X-ray crystallography or conventional whole-protein NMR, we site-specifically target this nearby Trp residue by substituting it into fluorinated Trp utilizing a Trp auxotrophic strain RF12[37–40] for $^{19}$F NMR measurement. The $^{19}$F chemical shift is hyper-responsive to different chemical environments[41] and because of the small van der Waals radius, $^{19}$F incorporation is generally biocompatible and structural nonperturbing[42,43]. As a result, an extrinsic $^{19}$F label of fluorinated Trp allows the quantification of the Trp conformation distributions free of the protein background NMR signals[44–50]. We first carried out NMR experiments on 5-fluorotryptophan (5FW) substituted wild type (WT) BLUF domain in AppA, SyPixD and OaPAC, since 5FW is the most frequently used as a $^{19}$F NMR probe among all fluorinated Trp amino acids[41]. Before delving further into the discussion, we conducted TA experiments to assess the impact of 5FW as well as 7-fluorotryptophan (7FW) (Supplementary Fig. 1). Remarkably, the kinetics at 760 nm revealed only minimal alterations with the same cross-species trend affirming that the central photocycles remained unaltered upon fluorotryptophan incorporation.

As shown in Fig. 2, AppA WT displays a single symmetrical $^{19}$F peak (denoted as Peak 1) even at low temperatures (Supplementary Fig. 2), suggesting that nearby W104 in AppA has a homogeneous conformation. Compared with the free 5FW signal located at −124.8 ppm higher field (lowest panel), the 5FW peak in AppA is significantly downshifted to −122.0 ppm (Supplementary Table 1). In the bulk water, the $^{19}$F chemical shift is sensitive to the ratio of $H_2O$ and $D_2O$ due to the different interaction between the fluorine and H/D atoms[51], and the $H_2O$/$D_2O$ experiment can be used to determine the solvent exposure of the NMR probe[52]. In the presence of 10%, 50%, and 90% $D_2O$, no shift is detected for AppA WT Peak 1, in contrast to the solvent-exposed free 5FW, which has an upfield shift of ~0.2 ppm (Supplementary Fig. 3, Supplementary Table 2). The above solution-NMR results for AppA indicate that the W104 is entirely buried inside the protein scaffold and

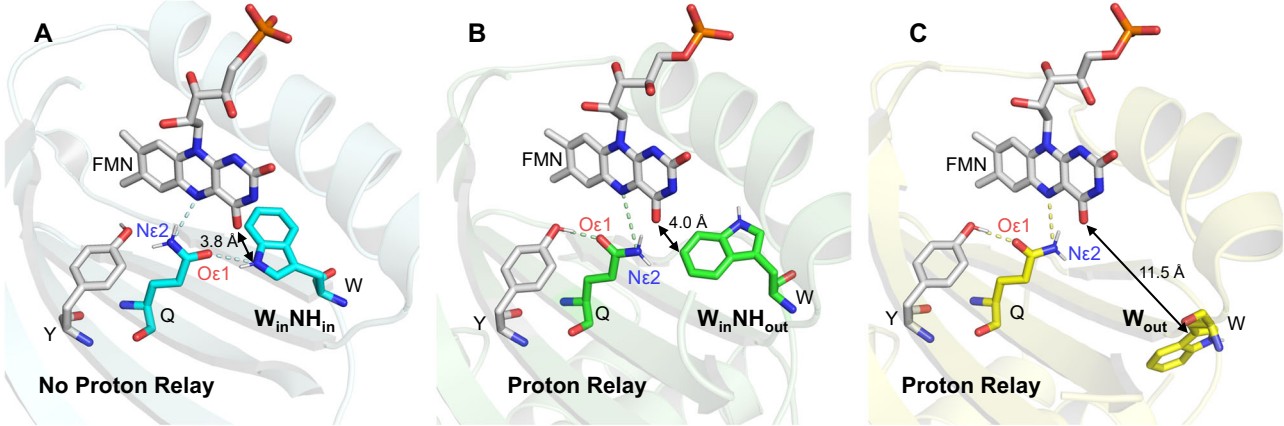

**Fig. 1 | Three active-site conformations, including the central FMN, Gln, and Tyr, as well as the nearby Trp in BLUF domains.** Three active-site conformations, including the central FMN (gray), Gln (Q, cyan/green/yellow), and Tyr (Y, gray), as well as the nearby Trp (W, cyan/green/yellow) in BLUF domains. $W_{in}NH_{in}$ configuration (**a**) is adopted from one X-ray structure of the AppA BLUF domain (PDB: 1YRX). A preferred no-proton relay geometry of Tyr and Gln is depicted according to previous free energy calculations[17]. $W_{in}NH_{out}$ (**b**) and $W_{out}$ (**c**) configurations are adopted from the two subunits of SyPixD tetramer (PDB: 2HFO), chain D, and chain B, respectively. Both favor a proton transfer geometry where Tyr OH is H bonded to Gln O$\varepsilon$1, and Gln N$\varepsilon$2H is H bonded to FMN N5[17,25]. Edge-to-edge distances (double arrows) between the nearby Trp and FMN are shown in each configuration. Possible proton transfer pathways are drawn in dashed lines.

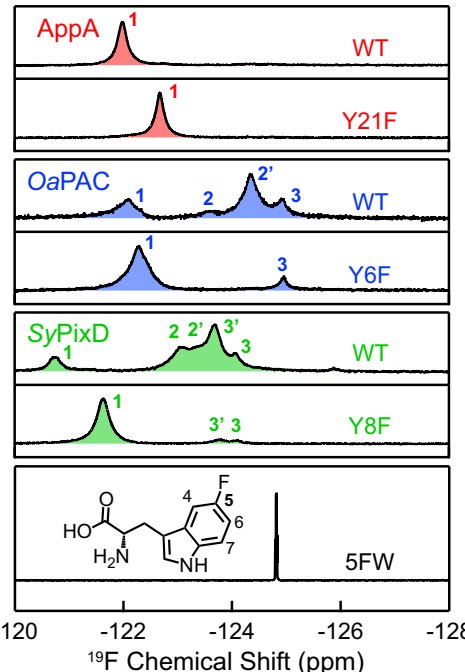

**Fig. 2 | $^{19}$F solution NMR results of 5FW labeled WT and YnF mutants of AppA, *Oa*PAC, and *Sy*PixD BLUF domains and 5FW alone in 10% D$_2$O at 298 K.** The $^{19}$F solution NMR peaks of AppA, *Oa*PAC, and *Sy*PixD BLUF domains are colored red, blue and green, respectively. Their resolved peaks are marked with numbers (1, 2/2′, 3/3′), denoting three classified configurations. Peak 1 has been assigned to the W$_{in}$NH$_{in}$ configuration, Peak 2 and 2′ have been assigned to subpopulations of the W$_{in}$NH$_{out}$ configuration, and Peak 3 and 3′ have been assigned to subpopulations of the W$_{out}$ configuration. The chemical structure of 5FW with atomic numbering is depicted in the lower panel.

is shielded from the solvent. In order to distinguish whether the buried peak is from W$_{in}$NH$_{in}$ or W$_{in}$NH$_{out}$, we designed a Y21F mutant eliminating the major electron and proton donor Y21, with W104 left as the only possible electron and/or proton donor for the excited FMN*. Here, we emphasize that the Y21F mutant is not photoactivatable and the central FMN-Gln-Tyr is no longer used for the PCET. As a result, we can deduce the configuration of Trp in Y21F from the resolved photo-chemistry using ultrafast TA spectroscopy in the next Section. As shown in Fig. 2, the single peak is upshifted from −122.0 ppm in WT to −122.7 ppm in Y21F, yet strikingly preserves a similar 0.20 ppm full width at half maximum (FWHM) (Supplementary Table 1). The similar FWHM suggests that both Peak 1 bands in WT and Y21F result from the same Trp configuration, although the chemical environment can be perturbed through the H-bond network via the central Gln upon the elimination of Tyr, explaining the NMR peak upshift.

In contrast to AppA WT, both *Oa*PAC and *Sy*PixD display multiple $^{19}$F NMR peaks (Fig. 2), showing the existence of conformational inhomogeneity in the ground-state ensembles in those two proteins. A similar pattern is observed in both WT NMR spectra, with one peak (denoted as Peak 1) in the downfield having different populations (28% for *Oa*PAC, 12% for *Sy*PixD) and several clustered peaks in the upfield. Most interestingly, after mutating the central Tyr into Phe (Y6F for *Oa*PAC, Y8F for *Sy*PixD), the conformation distributions altered in a similar manner, with a major peak (86% for *Oa*PAC, 90% for *Sy*PixD) slightly upshifted away from Peak 1 in WT and some minor peaks in the upfield. Similar to AppA, the $^{19}$F NMR linewidths for Peak 1 in *Oa*PAC and *Sy*PixD remain essentially unchanged upon Tyr mutation (FWHM: *Oa*PAC 0.43 to 0.41 ppm, *Sy*PixD 0.26 to 0.28 ppm, as shown in Supplementary Table 1). These akin chemical shifts for Peak 1, the

congruous peak shift patterns from WT to YnF (i.e., Y21F for AppA, Y6F for *Oa*PAC, Y8F for *Sy*PixD), and the retention of the linewidths together insinuate that Peak 1 in all three proteins result from a homologous conformation, and such a conformation becomes predominate upon Tyr mutation. In later sections, using ultrafast TA spectroscopy and advanced target analysis, we further identify the precise configuration of Trp for Peak 1, such as W$_{in}$NH$_{in}$ or W$_{in}$NH$_{out}$.

**Ultrafast spectroscopy for YnF shows a W$_{in}$NH$_{in}$ configuration**
As shown in Fig. 1a, in the W$_{in}$NH$_{in}$ configuration, previous free energy calculations suggested that instead of forming an H bond with Tyr OH on the left, the Gln Oε1 is prone to orient towards the Trp NH on the right (Fig. 1). Hence in YnF, when a W$_{in}$NH$_{in}$ configuration is adopted, after Trp donates an electron to the FMN*, a proton may transfer from the charge-separation species WH$^{·+}$ to the intermediating Gln, and possibly further relay to FMN$^{·-}$ if Gln Nε2H is within H-bond distance to FMN N5. Alternatively, for the W$_{in}$NH$_{out}$ conformation (Fig. 1b), Trp is within 3.7–4.0 Å edge-to-edge distance to FMN, allowing the occurrence of photo-induced electron transfer (ET). However, a proton relay to FMN N5 is prohibited due to the lack of a continuous proton wire in this configuration. Last, in the W$_{out}$ configuration, Trp is too distant (>10 Å, Fig. 1c) to quench FMN* by ET and/or proton transfer (PT). Hence, in the W$_{out}$ configuration, FMN* presumably undergoes a nanosecond intersystem crossing (ISC) to form $^3$FMN, similar to what happened for the YnF/WnF (i.e., Y21FW104F for AppA, Y6FW90F for *Oa*PAC, Y8FW91F for *Sy*PixD) double mutant[14,53]. The above analysis suggests that each Trp configuration corresponds to distinct photochemistry in YnF, and thus, the Trp configuration attributed to Peak 1 can be deduced from the TA results.

Femtosecond broadband transient UV/Vis absorption experiments were conducted on the YnF mutants of all three species with a pump wavelength of 480 nm in both H$_2$O and D$_2$O conditions. The raw two-dimensional (2D) contour map in H$_2$O is shown in the upper panel of Fig. 3a–c (D$_2$O TA data shown in Supplementary Fig. 4). Prominent FMN* characteristics are observed at the outset of the 2D contour maps for all three species, with excited-state absorption at above 700 nm and below 400 nm, simulated emission centered at 550 nm, and ground-state bleaching around 450 nm (see Supplementary Fig. 5 for reference spectra). In order to unveil the underlying intermediates masked by the intensive FMN* signals, we subtracted the FMN* contribution using the data analysis methodology developed by Tahara[54,55] and our group[14] (referred to as sub-2D TA maps as shown in the lower panels in Fig. 3a–c). The sub-2D TA maps exhibit diverse evolutions on initial inspection, yet a close examination reveals a congruous emergence of a charge-separated (CS) state at the early delay time (also shown in Fig. 3d–f, blue lines), in which the positive absorption in the UV region along with the peak around 510 nm likely arises from FMN$^{·-}$, whereas the broad absorption peak at 600-–650 nm likely arises from WH$^{·+}$, resembling the CS intermediates in previously reported FMN-Gln-Trp[56] and FMN-Glu-Trp[57] mutants. Intriguingly, as shown in Fig. 3a–c, the CS intermediates evolve into distinct PT states in the three BLUF domains. A later-time spectral snapshot cut from the sub-2D TA map of *Oa*PAC (Fig. 3f, red line) exhibits a characteristic feature resembling the di-radical (DR) pair FMNH$^{·}$/W$^{·}$ in previously reported FMN-Gln-Trp mutant (Supplementary Fig. 5)[56], with a negative absorption around -370 nm and a broad band at 510 nm to 600 nm attributed to FMNH$^{·}$ and a 540-nm bump corresponding to W$^{·}$. It can be deduced that *Oa*PAC Y6F proceeds with a photo-induced forward ET from Trp to FMN* to form the CS state, followed by a double proton transfer mediated by Gln to form the DR pair FMNH$^{·}$/W$^{·}$ (see Supplementary Fig. 6 for the scheme), which is consistent with previous TRIR results[24]. For AppA Y21F, the two selected spectral snapshots (Fig. 3d) at early and later time delays from the sub-2D map show an evident shifting pattern that the broad 650 nm peak evolved into a

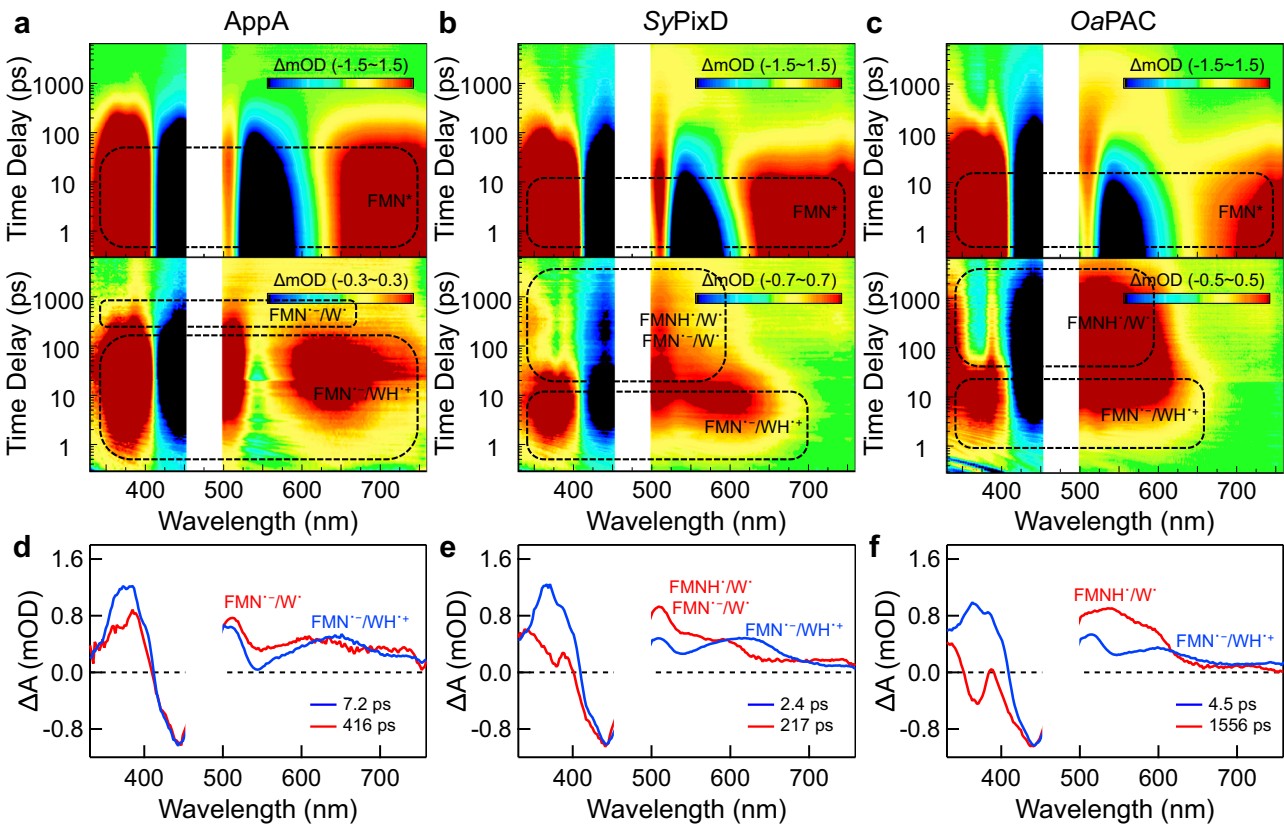

**Fig. 3 | Transient absorption two-dimension (2D) contour maps. a–c** Transient absorption 2D contour maps for AppA Y21F (**a**), *Sy*PixD Y8F (**b**), and *Oa*PAC Y6F (**c**) in H$_2$O buffer. The upper panel shows the original TA 2D spectra and the lower panel shows the 2D spectra after subtracting the contribution from FMN$^\bullet$ component. Detected transient species are denoted with dashed circles with internal annotations representing possible intermediates (FMN$^\bullet$, FMN$^{\bullet-}$/W$^\bullet$, FMN$^{\bullet-}$/WH$^{\bullet+}$, FMNH$^\bullet$/W$^\bullet$). **d–f** Scaled transient absorption spectral (red and blue lines) snapshots at selected time delays after FMN$^\bullet$ subtraction for AppA Y21F (**d**), *Sy*PixD Y8F (**e**), and *Oa*PAC Y6F (**f**) are shown to uncover the possible intermediates.

weak band ~600 nm, accompanied by an absorption abatement in the UV region, implying the deprotonation of WH$^{\bullet+}$ into W$^\bullet$ (see Supplementary Fig. 5 for reference spectra of WH$^{\bullet+}$ and W$^\bullet$) while preserving the spectral profiles of FMN$^{\bullet-}$ with a positive UV absorption and a characteristic 510-nm band. Hence, we attribute this transient species to the single PT state or the proton relay (PR) intermediate, identified as FMN$^{\bullet-}$/QH$^+$/W$^\bullet$, manifesting the status when the transferred proton from WH$^{\bullet+}$ is lingered on the Gln bridge without the occurrence of a second proton transfer from QH$^+$ to FMN$^{\bullet-}$. Most interestingly, for *Sy*PixD Y8F, a later-phase snapshot (Fig. 3e, red line) exhibits features from both the PR intermediate and the DR state, where the spectrum can be decomposed into a linear combination of the red lines in Figs. 3d and 3f (Supplementary Fig. 7). The above analysis suggests that despite the ostensible spectral differences, a unified picture can be deduced: after a photo-induce ET to generate the CS state, the three YnF mutants experience either a one-step (i.e., PT1, from the nearby Trp Nε1 to the central Gln) or a two-step PT (i.e., PT1 and PT2, from the central Gln Nε2 to the FMN N5) (see Supplementary Fig. 6 for the scheme), resulting in different populations of the PR intermediate and the DR state. The above TA results in three species unambiguously demonstrate that all three YnF mutants adopt a predominant W$_{in}$NH$_{in}$ configuration rather than W$_{in}$NH$_{out}$.

We then performed target analysis employing a bidirectional step-wise PT model, with the results shown in Fig. 4. We used a stretched function $f(t) = Ae^{-(t/\tau)^\beta}$ to represent the multi-phasic dynamics[14,58,59], in which $A$ is the amplitude, $\tau$ is the lifetime, and $\beta$ is the stretched parameter. To obtain the above parameters in the elementary steps, seven wavelengths were selected to fit the data globally (fitting details are provided in the Supplementary Methods

in Supplementary Information). The acquired lifetimes, stretched parameters $\beta$, and calculated averaged lifetimes are listed in Supplementary Table 3. The 2D TA maps were decomposed into the Species-Associated Differential Spectra (SADS) and species-associated kinetic traces (Fig. 4, middle and lower panels; D$_2$O target analysis results in Supplementary Fig. 8). The resolved SADS for FMN$^\bullet$, the CS state (FMN$^{\bullet-}$/WH$^{\bullet+}$) and $^3$FMN are consistent across the three species and agree with previously reported spectra (Supplementary Fig. 5)[14,56,60]. The main difference between the three lies in the appearance of the PR or DR intermediates, which can be comprehended from the resolved rates for the elementary steps, wherein PT2 plays an essential role. For example, in AppA Y21F, PT2 is too slow to occur within the TA time window, resulting in the absence of a DR intermediate. In *Oa*PAC Y6F, PT2 is too fast, preventing the accumulation of PR intermediate populations for detection, and thus, we can only resolve the total timescales for forwarding or reversing PT1 and PT2 together. On the other hand, in *Sy*PixD Y8F, fPT2 (608 ps) occurs neither too fast (slower than fPT1) nor too slow (within our 7-ns time window), allowing the sufficient accumulation of both PR and DR intermediates and all four PT steps can be dissected. The PT rate is highly sensitive to the H-bond distance $d$ which is exponentially related to the proton transfer rate $k_{PT}$ via Eq. (1)[61,62], where $k_{PT}^0$ are the PT rate constants at the equilibrium distance $d_0$. The exponential factor $\alpha$ for the proton transfer is typically 25–35 Å$^{-1}$ [61], and small distance fluctuations of 0.2 Å strongly affect the $k_{PT}$ in 3 orders of magnitude change. Hence, the difference in the PT2 rates observed here may reflect subtle distance variations in the H-bond networks across the species, indetectable by X-ray crystallography due to limited structural resolution, although we cannot exclude the

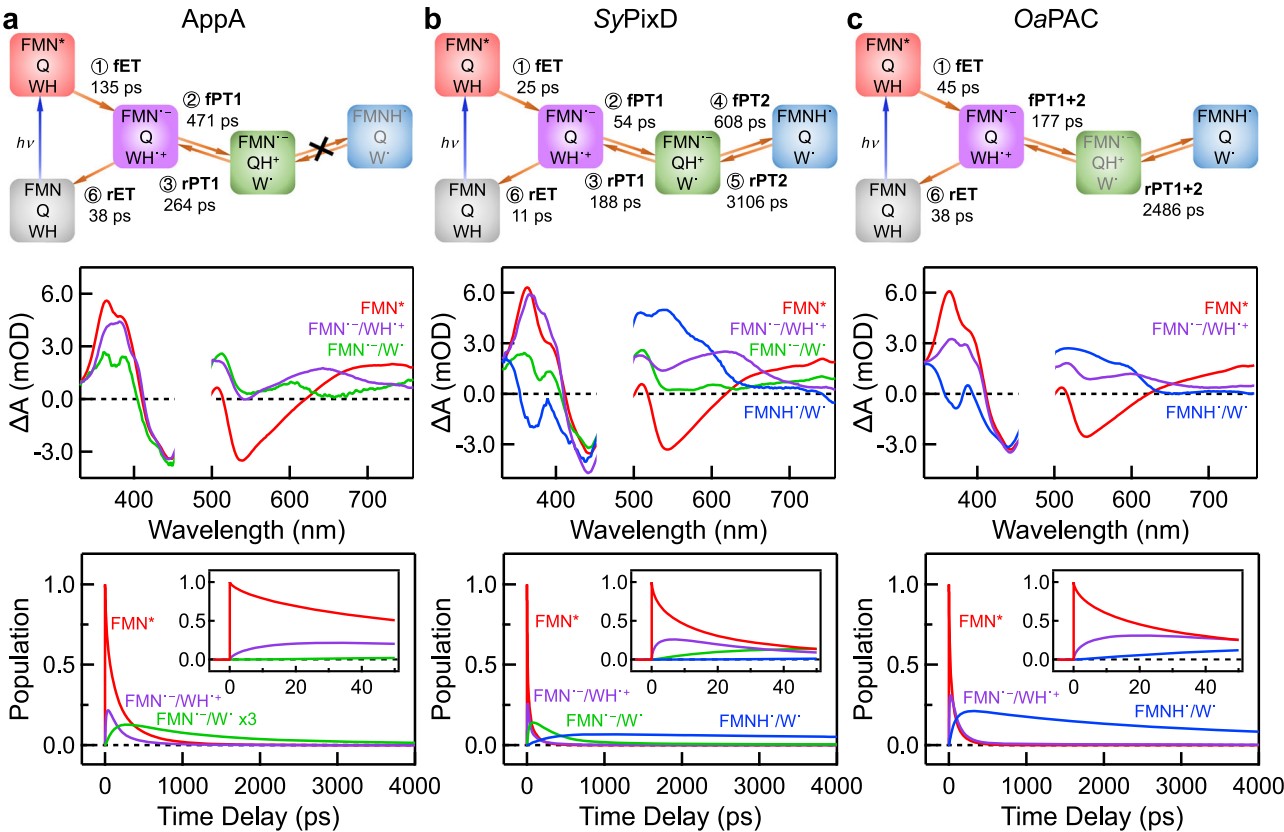

**Fig. 4 | Target analysis results of YnF mutants in H₂O buffer. a–c** columns show the target analysis results of AppA Y21F (**a**), *Sy*PixD Y8F (**b**), and *Oa*PAC Y6F (**c**) in H₂O condition, respectively. The upper panels illustrate the proposed kinetic models, single rocking (**a**) and double rocking models (**b, c**), for the photoreaction cycle. The arrows depict pathways with fitted averaged lifetimes, while the symbol "×" in (**a**) indicates that PT2 is inaccessible within our time window for the AppA Y21F mutant. In the middle and lower panels, the species-associated differential spectra (SADS) and corresponding kinetic traces are presented; FMN⁺ is depicted in red, FMN˙⁻/WH˙⁺ in purple, FMN˙⁻/W˙ in green, and FMNH˙/W˙ in blue. The inserts are the initial dynamics up to 50 ps.

possible effects of the local electrostatics of each BLUF domain protein scaffold[20,63–65].

$$k_{PT} = k_{PT}^0 \exp\left[-\alpha(d - d_0)\right] \quad (1)$$

Besides the predominant signals arising from the $W_{in}NH_{in}$ configuration, we captured additional minor ³FMN signals that emerged from the $W_{out}$ configuration, which is estimated to be ~4% and ~2% in *Sy*PixD and *Oa*PAC, respectively (Supplementary Fig. 9). Hence we attribute the minor Peak 3 in ¹⁹F NMR to $W_{out}$, with a proportion of 10% in *Sy*PixD and 14% in *Oa*PAC (Fig. 2 and Supplementary Table 1). The minor discrepancy in population distribution from TA and NMR experiments could result from the perturbation of the fluorinated NMR label to the conformation energetics.

### Identification of the $W_{out}$ and $W_{in}NH_{out}$ Configurations

After we had assigned all Peak 1 in the solution NMR results for WT and YnF (Fig. 2) to $W_{in}NH_{in}$, we revisited Fig. 1 and questioned the origin of those clustered peaks in the upfield. For *Oa*PAC Y6F, Peak 3 can be attributed to the $W_{out}$ configuration because Peak 3 aligns well with the peak that arises from the free 5FW (Fig. 2), the same configuration as in the full-length *Oa*PAC X-ray structure where the Trp sidechain is entirely solvent-exposed and located far from the active site (-12 Å, Supplementary Fig. 10). Such a Trp configuration is presumably free from the influence of the Y6F mutation. Hence, the chemical shift of Peak 3 remains unaltered in WT and Y6F. This assignment is in agreement with our TA results, where ~2% minor $W_{out}$ configuration is present (Supplementary Fig. 9), corresponding to Peak 3 in Y6F. For *Sy*PixD, since we also detected ~4% minor $W_{out}$ configuration in TA, we

temporally attributed Peak 3 and 3′ in Y8F as $W_{out}$, which becomes a predominant configuration in WT as the area for Peak 3 and 3′ in WT has significantly incremented. Since there is also a minor population with two distinct $W_{in}NH_{out}$ configurations in the *Sy*PixD X-ray crystal structures (Supplementary Fig. 11), we tentatively assign the unattained Peak 2 and 2′ in *Sy*PixD as arising from $W_{in}NH_{out}$. Nonetheless, in 5FW experiments for *Oa*PAC and *Sy*PixD, subpopulations for Peak 2 are clustered with those for Peak 3, and therefore, further experiments are required for a reliable quantification for both.

Next, we screened other fluorotryptophans, including 4-fluorotryptophan (4FW), 6-fluorotryptophan (6FW), and 7FW, by incorporating each of them into AppA, *Sy*PixD, and *Oa*PAC for ¹⁹F NMR detection. Among them, the 7FW NMR shows the best performance in distinguishing the three classified Trp conformations (Fig. 5). As depicted in Fig. 5, Peak 3 in *Oa*PAC (−134.2 ppm) and *Sy*PixD (−134.2 ppm) aligns well with free 7FW (−134.2 ppm). We conducted D₂O dependent experiment in *Sy*PixD (Supplementary Fig. 12), Peak 3 as well as free 7FW upshifted upon D₂O increase, yet the positions of Peak 1 and Peak 2 are independent of the D₂O percentage, which indicates both Peak 1 and 2 correspond to Trp conformations buried within the protein scaffold. At this point, we assign Peak 1 to $W_{in}NH_{in}$, Peak 2 to $W_{in}NH_{out}$ and Peak 3 to $W_{out}$. Further quantum chemical calculations[66,67] and solvent paramagnetic relaxation enhancement experiment[68] may provide additional support for these assignments. The ratios for Peak 1, 2, and 3 in the 7FW experiment agree remarkably with those in the 5FW measurement, reflecting the accurate percentages of $W_{in}NH_{in}$, $W_{in}NH_{out}$, and $W_{out}$ in the three BLUF domains, respectively. To recapitulate, 5FW and 7FW NMR results show that AppA contains a uniform $W_{in}NH_{in}$ conformation, *Oa*PAC has a

predominant $W_{in}NH_{out}$ configuration (57–61%), and *Sy*PixD has a major $W_{out}$ configuration (48–50%) as summarized in Supplementary Table 1. Interestingly, within the $W_{in}NH_{out}$ and $W_{out}$ configurations, both 5FW and 7FW NMR results show the existence of subpopulations corroborating the conformation heterogeneity captured in X-ray structures (Supplementary Fig. 11 and Supplementary Fig. 13). 5FW NMR performs better in resolving those subpopulations, in which we label with 2 and 2′ for $W_{in}NH_{out}$ as well as 3 and 3′ for $W_{out}$, whereas 7FW NMR shows an asymmetry in the lineshapes of Peak 2 in *Oa*PAC and Peak 3 in *Sy*PixD.

## FMN* quenching dynamics across the species

It has been an un-settled problem[15,16] why the primary FMN* quenching rates vary significantly across different BLUF domains, with a ubiquitous multi-phasic behavior. Previous researchers have utilized non-natural amino acids to substitute the central Tyr for redox tuning and found that the kinetics changes cannot be explained using a pure ET quenching mechanism[15,22]. Moreover, recent simulation work has suggested that the quenching rate is determined by the Tyr-Gln H-bond configuration, denoted as proton relay or no proton relay geometry, as shown in Fig. 1. The preferred orientation of Gln is determined by the energetics of the protein matrix environment,

manifested by the Trp configurations, such as $W_{in}NH_{in}$, $W_{in}NH_{out}$, and $W_{out}$[17,25].

Now that we have experimentally quantified the Trp configuration distributions in a cross-species manner in the above sections, we revisit the FMN* decay kinetics in order to find the correlation between the structural heterogeneity and the multi-phasic behavior. Here, focusing on the photocycle of the functional FMN-Gln-Tyr triad, we expressed and purified all three BLUF domain samples with the nearby interfering Trp mutated into ET-inert Phe and performed TA results retrieving the kinetics at 760 nm (Fig. 6). The 760 nm single wavelength kinetics for the three WnF (i.e., W104F for AppA, W90F for *Oa*PAC, W91F for *Sy*PixD) BLUF domains were analyzed using two approaches. First, all kinetics can be fitted with the conventional multi-exponential decay model (Supplementary Fig. 14 and Supplementary Table 4), resulting in an averaged lifetime trend of *Sy*PixD (105 ps) <*Oa*PAC (224 ps) <AppA (935 ps). However, the conventional exponential decay model cannot dissect the kinetic contribution of the ISC process from FMN* to the triplet state $^3$FMN. The alternative approach is to invoke the stretched parameter $\beta$ to describe the muti-phasic FMN* decay and a parallel kinetic model to include the ISC process, i.e., $f(t) = -Ae^{-(t/\tau_{fPCET})^{\beta_{fPCET}}} - Ae^{-(t/\tau_{ISC})}$. The results are shown in Supplementary Table 5, with the average lifetimes *Sy*PixD (201 ps) <*Oa*PAC (344 ps) <AppA (3036 ps), and the stretch parameters $\beta$ *Sy*PixD (0.35) < *Oa*PAC (0.50) < AppA (0.71). As the $\beta$ value in PCET reactions reflects the coupling with solvation and/or the H-bond fluctuations[14], a large $\beta$ value of 0.71 indicates that AppA has the most homogeneous H-bond network; on the contrary, the H-bond interaction in *Sy*PixD is the most heterogeneous, resulting in an unprecedented small $\beta$ value of 0.35. The relation between kinetics and structural heterogeneity will be further elaborated in the "Discussion".

## Discussion

Since Tyr is the key electron and proton donor, YnF is one of the most intensively studied mutants by ultrafast spectroscopists[19,24,53–71]. Using transient mid-IR spectroscopy, the Lukacs, Tonge, and Meech groups have detected both FMN•− and FMNH• intermediates in *Oa*PAC Y6F[24], captured the infrared signature for FMN•− within the 1-ns time window in *Sy*PixD Y8F[19] and identified the existence of FMN•− in AppA Y21S[69], wherein the overall observation matches with our TA results. In this work, combining $^{19}$F NMR and advanced target analysis, we provide a coherent cross-species picture for such diverse photochemistry. First, upon mutation, we found that $W_{in}NH_{in}$ becomes the predominant configuration across all three species (Fig. 2 and Supplementary Table 1). After abolishing the interaction between Tyr OH and Gln C = Oε1, the Gln sidechain is allowed to rotate freely, and Gln C = Oε1 becomes available for H-bonding with Trp Nε1H, thereby stabilizing $W_{in}NH_{in}$. Second, when in $W_{in}NH_{in}$, all three YnF mutants experience a photo-induced charge separation between FMN* and Trp,

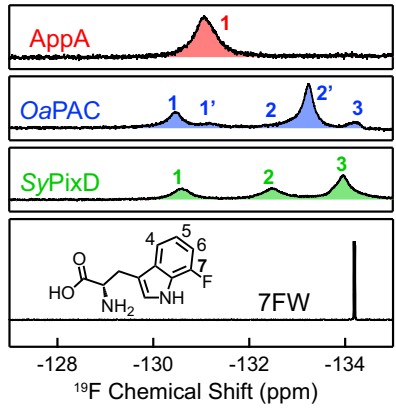

**Fig. 5 | $^{19}$F solution NMR results of 7FW labeled WT samples of AppA, *Oa*PAC, and *Sy*PixD BLUF domains and 7FW alone in 10% D$_2$O at 298 K.** The $^{19}$F NMR spectra of AppA, *Oa*PAC, and *Sy*PixD BLUF domains are shown in red, blue, and green, respectively. Their resolved peaks are marked with numbers (1/1′, 2/2′, 3), denoting three classified configurations. Peaks 1 and 1′ correspond to subpopulations of the $W_{in}NH_{in}$ configuration, Peaks 2 and 2′ relate to subpopulations of the $W_{in}NH_{out}$ configuration, while Peak 3 represents the $W_{out}$ configuration. The chemical structure of 7FW, along with atomic numbering, is illustrated in the lower panel.

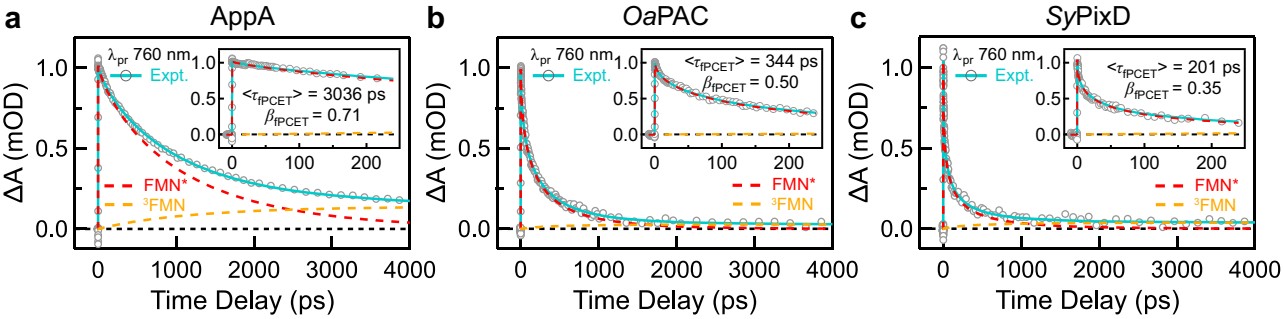

**Fig. 6 | Kinetic transient cuts at 760 nm wavelengths and fitting results. a–c** A stretched exponential function and a parallel kinetic model were adopted to analyze the kinetic traces at a selective probe wavelength of 760 nm for AppA W104F (**a**), *Oa*PAC W90F (**b**), and *Sy*PixD W91F (**c**) in H$_2$O buffer. The raw data is presented as symbols (○), while the fitted traces are shown as straight lines. Decomposed dynamics of FMN* (red dashed lines) and $^3$FMN (orange dashed lines) are shown up to 4000 ps with insets up to 250 ps.

followed by either a single proton rocking (PT1) between TrpH$^{\cdot+}$ N$\epsilon$1H and Gln C = O$\epsilon$1 (AppA Y21F) or a double proton rocking with a second PT2 between GlnH$^{+}$ N$\epsilon$2H and FMN$^{\cdot-}$ N5 (*Oa*PAC Y6F and *Sy*PixD Y8F), hence forming different populations of the PR intermediate (FMN$^{\cdot-}$/ QH$^{+}$/W$^{\cdot}$) and the DR pair (FMNH$^{\cdot}$/W$^{\cdot}$). The PR intermediate is largely overlooked in previous YnF and WT studies because the proton relays in BLUF domains are through a compact H-bond network and are naturally perceived as concerted[15,72]. In a recent report, we invoked the concept of a step-wise proton translocation[73] in the BLUF domain and detected the reverse PR intermediate in FMN-Glu-Trp with a nature of FMNH$^{\cdot}$/E-COO$^{-}$/WH$^{\cdot+}$ [57], where the deprotonation from the Glu-COOH to FMN$^{\cdot-}$ (PT2) occurs ahead of PT1 because of the acidity of Glu-COOH (p$K_a$ ~ 4.4[74]). In the current work, with the sidechain p$K_a$ ~ 19.6[75], the bridging Gln must accept a proton first from the TrpH$^{\cdot+}$ (PT1), before donating a proton to the FMN$^{\cdot-}$ (PT2). Here, we demonstrate the existence of the forward PR intermediates in AppA Y21F and *Sy*PixD Y8F, with the nature of FMN$^{\cdot-}$/QH$^{+}$/W$^{\cdot}$. Most importantly, we find that the major cross-species discrepancy originates from PT2, the occurrence of which is prohibited in AppA and is accelerated in *Oa*PAC, resulting in different populations of PR and DR intermediates. Since the bridging Gln is also involved in the functional WT photocycle, we believe that the endeavor in dissecting the step-wise proton relay will facilitate a cross-species elucidation of the photoactivation mechanism.

There have been long-standing debates regarding the Trp conformations due to the diverse and contradictory results in X-ray crystal and NMR reports[17,25,27–31,76]. Our site-specific $^{19}$F solution NMR discloses that in each BLUF domain of three species, there is a distinct Trp configuration distribution. For the dark-state AppA BLUF domain, our $^{19}$F NMR results demonstrate a homogenous W$_{in}$NH$_{in}$ configuration. Our conclusion in AppA largely agrees with prior solution NMR observations, as the NMR ensembles in 2BUN contain mostly W$_{in}$NH$_{in}$ conformations[76], yet contradicts the X-ray crystallographic results, where the majority of AppA X-ray crystal structures (2IYI, 2IYG, 4HH0, 4HH1) suggest a W$_{out}$ conformation with only one outlier (1YRX) revealing W$_{in}$NH$_{in}$ [27–29]. Our results also corroborate with previous free-energy simulations, suggesting that W$_{in}$NH$_{in}$ is thermodynamically preferred in the ground state for AppA WT[17]. Our solution $^{19}$F NMR measurements on *Oa*PAC and *Sy*PixD both reveal three disparate configurations W$_{out}$, W$_{in}$NH$_{in}$, and W$_{in}$NH$_{out}$, only with different proportions, showing a more prominent heterogeneity than in previous X-ray structures. For instance, in the reported two *Sy*PixD X-ray structures[30], there are 3 subunits with W$_{in}$NH$_{out}$, and the other 17 subunits have W$_{out}$ present in the two protein tetramer; in *Oa*PAC full-length X-ray structures, only the W$_{out}$ configuration has been discovered[31]. Interestingly, previous simulation work has predicted the possible existence of W$_{in}$NH$_{in}$ in *Sy*PixD from free energy profile calculations[25], which is corroborated by our $^{19}$F NMR experiments. In summary, our site-specific $^{19}$F NMR results demonstrate that X-ray structures are subject to the crystal lattice confinement[25,32], which results in an enhanced sampling of the W$_{out}$ configurations and thus cannot accurately reflect the actual solution status. In particular, the W$_{in}$NH$_{in}$ configuration exists in all three BLUF domains, yet has never been captured in X-ray structures of *Sy*PixD or *Oa*PAC.

According to previous free energy calculations by the Hammes–Schiffer group, when the W$_{in}$NH$_{in}$ conformation is favored, Tyr and Gln are prone to orient in a so-called no proton relay configuration[17]; for the W$_{in}$NH$_{out}$ and W$_{out}$ conformations, a proton relay configuration between Tyr and Gln is thermodynamically preferred[25]. In this work, since we have experimentally quantified Trp conformation distributions using $^{19}$F NMR, a correlation between the active-site conformations and the FMN$^{*}$ decay dynamics can be sorted out. First, structural heterogeneity is not the only source for multiphasic behavior. Take AppA for instance, $^{19}$F NMR reveals a homogeneous structural distribution, yet the FMN$^{*}$ decay shows a multi-

exponential decay with the stretch parameter $\beta = 0.70$, which might arise from the dynamic heterogeneity, i.e., the coupling of ET with the rotational motions of the Tyr and Gln side chains to switch from no proton relay to proton relay configurations, and/or the coupling between ET and the solvation dynamics. Second, the structural heterogeneity, also termed static heterogeneity, does contribute because more diverse distributions (*i.e.*, more complicated NMR lineshapes) in *Oa*PAC and *Sy*PixD are correlated with smaller $\beta$ values (0.50 for *Oa*PAC and 0.35 for *Sy*PixD, Fig. 6), compared to AppA. Third, our results reveal that a higher W$_{in}$NH$_{in}$ proportion [AppA (100%) > *Oa*PAC (28 ~ 35%) > *Sy*PixD (12 ~ 23%) (Supplementary Table 1)] corresponds to a longer quenching lifetime [AppA (3036 ps) > *Oa*PAC (344 ps) > *Sy*PixD (201 ps) (Fig. 6)]. The underlying picture is that in W$_{in}$NH$_{in}$, the H-bonding pattern of the predominant no proton relay configuration[17] prohibits a PT-coupled ET reaction from Tyr to FMN$^{*}$, and the protein requires extra time to sample a less energy-favorable proton relay configuration. Together, we emphasize that the rate of FMN$^{*}$ decay is determined by the concerted PCET reaction rather than by a pure ET reaction and thus is highly sensitive to the Tyr-Gln H-bond pattern, i.e., to what extent it is facile for an efficient proton transfer. Additionally, it is important to recognize that other physical factors may also play a role. These factors encompass the electrostatics in the active site[20], the redox potential of flavin and Tyr[15], the distances separating the electron donor and acceptor[20], as well as the p$K_a$ values of Tyr[16].

In this work, using the site-specific $^{19}$F NMR combined with ultrafast spectroscopy, we addressed the debate regarding the structural heterogeneity of the sub-conserved Trp in three BLUF domains (AppA, *Oa*PAC, and *Sy*PixD) in the aqueous solution. For AppA, the nearby Trp adopts a 100% W$_{in}$NH$_{in}$ conformation in the ground state. In contrast, both *Oa*PAC and *Sy*PixD sample three different Trp conformations, namely W$_{in}$NH$_{in}$, W$_{in}$NH$_{out}$, and W$_{out}$, only with different percentages. Upon the Y mutation, the W$_{in}$NH$_{in}$ conformation also becomes predominant in *Oa*PAC and *Sy*PixD, where we captured both the one-step proton transfer PR intermediates and the two-step proton transfer DR pairs in the photocycles, consistent with the molecular picture that the Trp N$\epsilon$1H forms an H-bond with the central Gln C = O$\epsilon$1 in W$_{in}$NH$_{in}$. Finally, we correlated the Trp conformation distributions to the FMN$^{*}$ quenching dynamics and found that a more prominent static heterogeneity results in a smaller fitting stretched parameter and a smaller proportion of the W$_{in}$NH$_{in}$ leads to a faster FMN$^{*}$ decay by thermodynamically sampling more "no proton relay" configurations. Our work provides compelling experimental evidence that the conformational heterogeneity in the ground-state ensemble plays a crucial role in determining the observed differences in the initial light-activated step in BLUF domains, primarily through the concerted PCET mechanism. These findings enhance our understanding of the functional implications of structural diversity in BLUF domains and shed light on the underlying mechanisms of their photochemical processes.

## Methods

### Mutant construction

Full-length AppA$_{1-450}$ and OaPAC$_{1-366}$ contain additional downstream domains. In light of the potential influence of downstream domains on the conformational distribution of Trp (Supplementary Fig. 15), which can render fluorinated Trp less suitable as a readout, we have constructed AppA$_{1-124}$ and OaPAC$_{1-102}$, with a focus on the BLUF domain. The wild-type plasmids were constructed by incorporation of the protein sequence from GenBank, i.e., ABA77707.1 (AppA), AFY83176.1 (*Oa*PAC), BAA18389.1 (*Sy*PixD) into a pET28a vector at the NdeI-SacI, NdeI-HindIII, NdeI-BamHI restriction sites, respectively. Also, we constructed mutants plasmids (i.e., AppA$_{1-124}$ W64F, Y21F/ W64F and W104F/W64F, *Oa*PAC$_{1-102}$ Y6F and W90F, *Sy*PixD$_{1-150}$ Y8F and W91F) based on the wild type plasmid by site-specific mutagenesis using QuickMutation site-directed mutagenesis kit (D0206S,

Beyotime). The primers used in this work are listed in Supplementary Table 6.

## Protein expression and purification

The 5FW and 7FW labeled protein samples, i.e., WT and YnF mutants of $OaPAC_{1-102}$, $SyPixD_{1-150}$, and $AppA_{1-124}$ BLUF domains, were expressed in tryptophan auxotrophy RF12 strain (a gift from Robert Gennis & Toshio Iwasaki, Addgene plasmid # 62077)[37–40]. The knockout of *trpA* and *trpB* genes makes the RF12 strain an optimal genotype for tryptophan derivatives labeling. *Oa*PAC and *Sy*PixD each contain a single Trp amino acid, yet AppA consists of a second Trp site (W64) besides the critical W104 in our interest. Hence, the second W64 in AppA BLUF is mutated to Phe to avoid being substituted by fluorinated Trp amino acids and complicating NMR signal patterns (AppA W64F is referred to as AppA WT for clarity throughout the main text). The amino acid sequences used for fluorotryptophan incorporation in the current study are listed in Supplementary Table 7, and the sequence alignment figure of WT is provided in Supplementary Fig. 16.

To express the labeled protein, the pET28a-BLUF expression plasmid[56] was transformed into the host strain RF12. The transformants were grown at 30 °C in 1 L of M9 minimal medium (3.0 g/L $KH_2PO_4$, 6.8 g/L $Na_2HPO_4$, 0.5 g/L NaCl, 1.0 mM $MgSO_4$, 1.0 g/L $NH_4Cl$, 0.01 mM $FeSO_4$, 0.1 mM $CaCl_2$, 0.5% glucose, 0.1% thiamine-HCl, and 50 mg/L kanamycin) containing L-Tryptophan until $OD_{600}$ reached 0.5. The cells were then harvested and washed with M9 medium and subsequently resuspended in M9 medium containing 0.5 mM L-Tyrosine. The resuspended cells were incubated at 30 °C for 30 min before adding 0.5 mM fluorinated tryptophan. When $OD_{600}$ reached ~0.7, 0.7 mM isopropyl-β-D-thiogalactopyranoside (IPTG) was added and the labeled protein was overproduced for 15–17 h at 18 °C under dark conditions. The unlabeled AppA and *Oa*PAC were expressed in *E. coli.* BL21(DE3) strain after induction using 0.7 mM IPTG in dark condition for 15~17 hours at 18 °C, while *Sy*PixD was expressed by leakage without IPTG for 30 h at 25 °C. The cells were harvested by centrifugation and stored at −80 °C. After sonication, the centrifuged supernatant was incubated with FMN for at least 45 min. We opt for FMN over FAD in order to eliminate the potential electron transfer pathway between lumiflavin and adenine within FAD. Then Ni-NTA Agarose columns and SDG-25 columns were used sequentially for protein purification[56]. The purified protein was quick-frozen using liquid nitrogen and was stored at −80 °C, in 100 mM NaCl, pH 8.0 50 mM $Na_2HPO_4/NaH_2PO_4$, 2 mM DTT, 5% glycerol buffer with a concentration of 200–300 μM.

## Mass spectrometry

For liquid chromatography-mass spectrometry, the purified protein samples were diluted to 50–100 μM. The LC-MS experiments were performed on Waters Xevo G2-XS-qTOF-MS (Waters, Medford, MA, USA) equipped with an electrospray ionization (ESI) source in conjunction with Waters ACQUITY UPLC I-Class Plus System. Separation and desalting were carried out on an ACQUITY UPLC Protein BEH C4 Column. Mass spectral deconvolution and the prediction of the molecular weight was performed using UNIFI software (version 1.9.4, Waters Corporation). The incorporation levels of tryptophan analogs are 70–98%, as shown in Supplementary Fig. 17.

## 19F NMR spectroscopy

For NMR measurement, the protein sample was exchanged into 25 mM NaCl, 12.5 mM $Na_2HPO_4/NaH_2PO_4$ (pH 8.0) buffer using SDG-25 column chromatography and then lyophilized into powder overnight and re-solubilized in 10% $D_2O$ (pD 8.0) as a lcoking agent[77], reaching a final concentration of 1.0~1.5 mM. Before loading into the NMR tube, the protein sample was centrifuged for 5 min at room temperature to remove any precipitates. All BLUF proteins were incubated in the dark for 30 min prior to NMR measurements.

Fluorine NMR spectra were recorded using Bruker Avance Neo 700 MHz NMR Spectrometer (Bruker, Germany) equipped with a TCI Cryoprobe. The BCU-I cooling unit was used to regulate the temperature from 278 K to 293 K. The 19F resonances were referenced to an external standard, trifluoroacetic acid (TFA), which was set at −76.55 ppm. To enhance the signal-to-noise ratio (S/N), each experiment was repeated and averaged 512 scans. The data analysis was performed using Topspin 4.1.4 and Python 3.11, and the chemical shifts and line widths are listed in Supplementary Table 1 and Supplementary Table 2.

## Ultrafast broadband transient absorption spectroscopy

The details on the ultrafast broadband transient absorption (TA) spectroscopy were described in our previous publication[14]. To summarize, the 480-nm pump beam was generated from an optical parametric amplifier (TOPAS-Prime, Spectra-Physics, USA) and a titanium/sapphire femtoseconds laser system (Solstice ACE, Spectral Physics, USA). The visible probe continuum (430–770 nm) and ultraviolet probe continuum (330–590 nm) were generated with a Sapphire crystal plate and a moving $CaF_2$ crystal plate, respectively, in a commercial TA spectrometer (Helios Fire, Ultrafast Systems, USA). The time window for the translational stage in the TA spectrometer is 7 ns. A 500 nm long-pass filter was placed before the detector for the visible region measurement, and a 450 nm short-pass filter was used for ultraviolet region measurement to screen the scattering light induced by the pump laser. Steady-state UV/vis absorption measurements were conducted on a TU-1810 UV/vis spectrometer (Beijing Purkinje General Instrument Co., Ltd., China) throughout the TA experiments to monitor the photo-damaging extent of the sample. For the TA measurement in $H_2O$, the concentration for the protein sample was ~200 μM, and the buffer was exchanged into 100 mM NaCl, 50 mM $Na_2HPO_4/NaH_2PO_4$ (pH 8.0). Before the TA measurement in the $D_2O$ condition, the protein sample went through three 5-h $D_2O$ (pD 8.5) incubation and lyophilization cycles to ensure a complete H/D exchange, as described in the previous publication[56].

## TA data analysis

The detailed data analysis methods are shown in Supplementary Methods. In brief, transient absorption 2D contour maps were first analyzed with sequential model fitting with the corresponding evolution-associated difference spectra (EADS) obtained by using the Glotaran 1.5.1. Then, the FMN* matrix was subtracted from the raw data using Matlab R21021a (ver. 9.10.0). After that, a physically-relevant model was constructed for target analysis for each protein, and the stretched exponential function was introduced to describe kinetic heterogeneity. In this process, the averaged lifetimes and β parameters were calculated by Matlab R21021a (ver. 9.10.0). Finally, the goodness of matrix deconvolution was determined by the following aspects. First, the SADS spectra agreement between $H_2O$ and $D_2O$. Second, a small residual matrix represents the difference between the experimental data and the fitted data. Third, the goodness of the fit of important kinetics at representative wavelengths. Fourth, the consistency of SADS with the proposed intermediates involved in the system.

## Reporting summary

Further information on research design is available in the Nature Portfolio Reporting Summary linked to this article.

# Data availability

The transient absorption experimental data and NMR experimental data have been deposited in the figshare database under the accession code [https://doi.org/10.6084/m9.figshare.24441943]. The BLUF structures used in this study were obtained from the Protein Data Bank (PDB) with accession codes 1YRX and 2HFO. Source data are provided as a Source Data file with this paper. Source data are provided in this paper.

## Code availability
The codes used in the data analysis are available from the corresponding authors upon request.

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

## Acknowledgements

B.D. acknowledges Dr. Yangyi Lu for stimulating discussions. The authors acknowledge Dr. Faming Lu for maintaining the laser facility. The authors sincerely acknowledge Prof. Toshio Iwasaki for developing the auxotrophic strains, sharing them, and helping us grow the RF12 strain. Funding support for B.D. from the National Key Research and Development Program of China (Grant No. 2020YFA0509700) and the National Natural Science Foundation of China (Grant No. 31971233) are acknowledged.

## Author contributions

B.D. and D.Z. conceived the concept. Y.Z., S.T., Z.C., Z.Z., J.H., X.K., S.Z. and T.Z. performed experimental studies. Y.Z., S.T., Z.C., X.K., B.W. and J.H. analyzed data. B.D. supervised the project. Y.Z., X.K. and B.D. wrote the original draft. Y.Z., S.T., B.D. and D.Z. reviewed & edited the paper.

## Competing interests

The authors declare no competing interests.
