## [Peer Review File · Nature Communications]

Origin of the Multi-phasic Quenching Dynamics in the BLUF Domains Across the SpeciesREVIEWER COMMENTS

Reviewer #1 (Remarks to the Author):

The manuscript by Zhou et al investigates how the structural heterogeneity of the semi-conserved trp affects the photochemistry mechanism across BLUF proteins. They incorporated specific ¹⁹F-Trp analogues in AppABLUF W64F, PixD and OaPACBLUF and using ¹⁹F NMR combined with ultrafast spectroscopy, the author quantified the Trp configuration distribution in the 3 mentioned BLUF proteins. According to the authors, the conformational heterogeneity in the ground-state plays a crucial role in determining the observed differences in the initial light-activated step in BLUF domains. Their results show that AppABLUF compared to PixD and OaPACBLUF does not form radical(s) because the Trp is in WinNHin configuration, preventing a PT-coupled ET reaction from Tyr to FAD*. OaPAC compared to PixD has slightly longer quenching lifetime due to higher proportion of WinNHin configuration. Even though the results of this manuscript are compelling, and the experiments are well designed, the authors cannot make the claim that they settled the long-term debate regarding the active-site heterogeneities in the dark-adapted state across different BLUF species. The Trp conformation distribution could be a possible explanation but there are other interactions around the flavin that could also explain the variation in photochemistry across BLUF proteins.

*If They want to make their finding more compelling for a Nature Communication publication, they need to include another BLUF protein that does not form radical like AppA and determine the % of the Trp configuration. For example, BlsA could be that other BLUF protein. Based on the authors' hypothesis, the trp configuration in BlsA is expected to be in WinNHin. Including another BLUF protein like BlsA can make their evidence stronger.

*Also authors should mention in their introduction all the Trp fluorescence studies done to investigate the Trpin/ Trpout configuration in AppABLUF and PixD. Karidi et al. 2020, Dragnea et al.2009, Yuan et al. 2011

*In this manuscript, the authors used only the BLUF domain (1-102) of OaPAC, omitting the effector domain could potentially affect the dynamic of Trp loop, how do you know that this is not what is causing the trp distribution difference between PixD and OaPAC in your NMR studies?

*Event though F has similar size as H, the electronegativity is different, thus the authors need to show that 5-FW and 7FW are not changing the behavior of the proteins and the fluorine is not controlling the conformation of the trp. The author should characterize 5-FW and 7-FW variants using TA to make sure the photocycle is not perturbed. They also can do the same experiment as in the supplemental table 3, KIE.

*minor comment- We use FAD and not FMN for BLUF protein.

Reviewer #2 (Remarks to the Author):

In this manuscript, Yalin Zhou et al. studied a blue light photoreceptor, BLUF domain, using ^{19}F NMR and femtosecond absorption spectroscopy. BLUF (blue light using flavin) photoreceptor is a photosensor that typically binds the FAD (flavin adenine dinucleotide) chromophore and functions through the photoinduced proton-coupled electron transfer reaction occurring in the chromophore-binding site. So far, there has been much debate about the position of tryptophan near the chromophore. The positions of tryptophan residue, such that it is oriented toward inside or outside the protein, can have significant influence on the PCET, but it has not been clearly determined in X-ray crystallographic studies and controversial.

In this study, they used the BLUF protein where the fluorinated tryptophan is incorporated in a site-specific manner, and the ^{19}F NMR data gave the clear insight about the positions of the tryptophan. Then, the ultrafast absorption measurement was used to observe the photoreaction, which provided not only the presence of the proton transfer between the tryptophan and chromophore but also the evidence for the hydrogen bond between them. I think that this study determined, in the clearest way, the heterogeneous positions/orientations of the important tryptophan residue in BLUF domain.

For this study, I have a couple of comments on their data and manuscript as below.

1. Section of "Ultrafast spectroscopy for YnF reveals a WinNHin configuration" or "Solution NMR with site-specific 5FW probes for AppA, OaPAC and SyPixD"

There is no description that explains that YnF mutant is not photoactivatable and/or FMN-Gln-Tyr is no longer used for the PCET.

2. References

I think that authors should cite the paper which first discovered the BLUF protein, such as "Masuda et al., Cell, 2002, 110, 613-23".

3. Section "Distinguishment and quantification for the Wout and WinNHout configurations. "

I agree that the peak 1 of ^{19}F NMR corresponds to WinNHin state, and the peak 3 corresponds to the Wout state, on the basis of the experimental data. However, the evidence to assign the peak 2 to WinNHout state seems to be weak. I think that some evidence will be necessary to support the assignment of the peak 2.

Is peak 2 dependent on the D₂O percent?

Or, if the quantum chemical calculations using QM/MM method or some reasonable model of the active site qualitatively reproduce the ^{19}F NMR peak positions of WinNHin and WinNHout, that would be sufficient for the assignment.

4. Figure 5

It is not clear why 10% D2O is used.

5. Introduction

Trpin is used in the introduction, but Win is used in other sections.

Reviewer #3 (Remarks to the Author):

Comments for NCOMMS-23-33858

The article entitled “Origin of the Multi-phasic Quenching Dynamics in the BLUF Domains Across the Species” by Zhou et al. investigated the multi-phasic relaxation of the photoexcited flavin cofactor in three BLUF domains, AppA, OaPAC and SyPixD, using transient absorption spectroscopy and ¹⁹F-NMR spectroscopy. The Blue light using flavin (BLUF) photoreceptors is an important family of photosensitive proteins, from which many photosynthetic organisms use blue light to control and regulate their growth and development. The mechanistic understanding of the photo-biophysics of BLUF domain is essential for further development of photo-controlled unit to biochemical applications. The crystal structures of BLUF domains suggested an ensemble of conformations could be sampled by the BLUF domains. However, it’s unclear how this conformation heterogeneity related to the complicate photochemical behaviors. The authors employed the highly conformation-sensitive ¹⁹F-NMR to identify the multiple states existed in the BLUF domain. The NMR peaks were assigned by solvent exposure of the ¹⁹F labeled residue and currently available crystal structures of AppA, SyPixD and OaPAC. Importantly, based on the correlation of the population of the NMR identified states and the relaxation properties of the photoexcited state of FMN* bound with BLUF domains, the authors found that the states identified from NMR is the origin of the multi-phasic relaxation behaviors of FMN*. Specifically, the authors demonstrated that the WinNHin conformation, which encodes a Tyr-Gln configuration and the related proton transfer pathways, is key for the central photoreactions. This work also demonstrated that combining ¹⁹F-NMR with ultrafast spectroscopy is powerful to study the multiple state dynamics of photosensitive proteins. The results are convincing and conclusions are clear. I believe this manuscript is suitable for publishing on Nat. Comm. But I have some comments and suggestions that may further increase the impact of the manuscript:

1. The manuscript is focusing on the question about the multi-phasic relaxation of FMN*. The authors may need to introduce with a few sentences why it is important. Excitation of the FMN eventually results in the BLUF domain change to other conformations. Does the FMN relaxation relate to the photoisomerizing properties of BLUF domain.
2. Related to question 1, the authors only gave the transient absorption spectra for the three proteins. How is the differences of the general photoswitching properties between AppA, SyPixD and OaPAC?
3. Although fluoro-substitution is generally recognized as a minimal perturbation to protein structure, it

affects the electron density distribution on the indole group of Trp. Thus, the fluoro-Trp may affect the photochemical process compared with the real wild type protein. If I understand correctly, it seems like the authors used ^{19}F -Trp labeled protein for ^{19}F -NMR, but use the none labeled protein for transient spectral measurement. Thus, at least one transient spectral measurement needs to be done using the ^5F -Trp labeled protein. Otherwise, the ^{19}F -NMR results might difficult to correlated with the transient spectra due to the unknown effect of fluoro- substitution.

4. The Y21F mutation abolishes the photo sensitivity of AppA (Biochemistry, 2003, 42(22):6726-34). However, the NMR spectrum of both WT and Y21F AppA showed only WinNHin state (Peak 1). How to explain this? The authors may need to give the rationale in the manuscript. The authors may consider to measure the ^{19}F -NMR spectrum of AppA under low temperature to see is the observed peak is a coalesced peak of multiple peaks due to a fast exchange in NMR time scale.

5. In line 93, the use of 2HFOD and 2HFOB is very confusing. Using "(PDB code: 2HFO), chain D and chain B, respectively" might be better.

6. The authors may need to give a note for YnF in the first time it appears.

7. Please make sure the full name is given in place where the abbreviation is used for the first time. For example, TA appeared on Line 163, but the full name is at Line 447.

8. Temperature has significant influence on the thermodynamic properties, i.e., state populations. However, I didn't see the temperature values anywhere in the manuscript.

9. More details of ^{19}F -NMR spectral processing may need to be added. For example, how did you do the peak deconvolution?

10. Assignment of the multiple ^{19}F -NMR peaks is very challenging. I would suggest using solvent PRE to see the differential solvent exposure of peak 1, 2/2' and 3/3'. This effect might be better observed by T1 relaxation measurement in the presence of caged gadolinium reagent such as Gd-DTPA-BMA. I would image that peak 3 and 3' should have fastest solvent PRE effect.

11. A supporting figure to show the sequence alignment of AppA1-124, OaPAC1-102 and SyPixD1-150 would be very helpful and allow readers to see the conservative residues quickly.

We appreciate that the reviewers are experienced experts in this field and their comments are quite insightful, which help us improve the quality of our manuscript. We have done our best to address the comments of the reviewers and have performed several additional experiments as suggested. We have included in the response letter a) ^{19}F NMR results of an AppA-type BLUF domain BlsA; b) the comparison of the TA results from F-W labeled and non-labeled samples; c) the comparison of the NMR results from *OaPAC* 1-102 and 1-126 samples; d) the D_2O dependent experiment to assign Peak 2 and e) the temperature dependence for the NMR spectra of the AppA samples. The changes made by the point-by-point responses to the reviewers' comments are as follows and we have revised the manuscript and supporting information accordingly.

The point-by-point responses to the reviewers' comments.

Reviewer 1:

We thank Reviewer 1 for stating that “the results of this manuscript are compelling and the experiments are well designed”. Besides, we agree with reviewer's comment “...there are other interactions around the flavin that could also explain the variation in photochemistry across BLUF proteins.” Therefore, we have revised some of our general statements on Page 20:

“Additionally, it's important to recognize that other physical factors may also play a role. These factors encompass the electrostatics in the active site²⁰, the redox potential of flavin and Tyr¹⁵, the distances separating the electron donor and acceptor²⁰, as well as the pK_a values of Tyr¹⁶.”

Sub-Section 1: Major comment:

1) *If They want to make their finding more compelling for a Nature Communication publication, they need to include another BLUF protein that does not form radical like AppA and determine the % of the Trp configuration. For example, BlsA could be that other BLUF protein. Based on the authors' hypothesis, the trp configuration in BlsA is expected to be in $W_{in}NH_{in}$. Including another BLUF protein like BlsA can make their*

evidence stronger.

Author reply:

We greatly appreciate the insightful comment from Reviewer 1. To strengthen the robustness of our findings, we conducted ^{19}F NMR measurements on another AppA-type BLUF protein, BlsA. Besides the nearby Trp W92, BlsA also includes an extra W78, and thus we use W78F mutant as a template for 5FW incorporation in order to get the signal from the nearby W92 only. The results are as shown in Figure C1 and are interpreted follows: BlsA displays a single symmetrical ^{19}F peak at -119.2 ppm in the lowest downfield region, indicating a predominant $W_{\text{in}}\text{NH}_{\text{in}}$ configuration similar to AppA. This finding aligns with the conclusion presented in our manuscript, further supporting the notion that a higher $W_{\text{in}}\text{NH}_{\text{in}}$ proportion corresponds to a longer quenching lifetime and the absence of intermediate radical.

Figure C1. ^{19}F solution NMR results of 5-fluorotryptophan (5FW) labeled BlsA (using W78F as a template), AppA, *OaPAC*, and *SyPixD* WT BLUF domains and 5FW alone in 10% D_2O at 298 K. Note there is an impurity resonance at -119.6 ppm. The result for BlsA is preliminary, and is only for scientific communications.

However, it is worth noting that the signal-to-noise ratio of this NMR measurement on BlsA is not as optimal as in other cases. The reason for this is the remarkably low

yield of soluble protein for BlsA W78F with our pET28 plasmid, which was only 0.4 mg/L, despite our extensive efforts, including scaling up to 7 L culture to obtain the necessary NMR results. Since the result for BlsA is preliminary, we decide not to include it in the current version of manuscript. Nonetheless, we are aware that BlsA is the only BLUF domain that exhibits a red-shifted UV/Vis absorption in the dark state (Tonge *et al.*, *J. Phys. Chem. Lett.*, **2014**, 5, 220), together with the ¹⁹F peak in the lowest downfield region among all four proteins, we believe that BlsA has an intriguing H-bond network in the active site that is worth further investigation.

2) *Also authors should mention in their introduction all the Trp fluorescence studies done to investigate the Trp_{in}/ Trp_{out} configuration in AppA_{BLUF} and PixD. Karidi et al. 2020, Dragnea et al.2009, Yuan et al. 2011.*

Author reply:

We thank Review 1 for this critical comment. Trp fluorescence is an important tool in studying the Trp conformations in the aqueous solution. We have added one sentence in Paragraph 1, Page 4 in the manuscript as follows, with the mentioned references cited. “Despite fluorescence measurements performed to interrogate the Trp conformations in aqueous solutions³³⁻³⁵, the relation between Trp conformation and the H bond network in the active site remains largely unknown.”

3) *In this manuscript, the authors used only the BLUF domain (1-102) of OaPAC, omitting the effector domain could potentially affect the dynamic of Trp loop, how do you know that this is not what is causing the trp distribution difference between PixD and OaPAC in your NMR studies?*

Author reply:

This is a highly insightful comment related to our experimental design. Both the full-length AppA₁₋₄₅₀ and OaPAC₁₋₃₆₆ contain additional downstream domains and we used AppA₁₋₁₂₄ and OaPAC₁₋₁₀₂, with a focus on the BLUF domain. The premise of our experimental design is to probe the H-bond network using the conformation of the nearby Trp as a readout. Hence, we need to reduce other factors that may also affect the

conformation of the nearby Trp, for instance the crystal lattice tends to increase the proportion of W_{out} as well as the interaction of effector domain in some cases. In fact, at the early stage of our project, we already conducted ^{19}F solution NMR experiments on two constructs: 5-fluorotryptophan (5FW) labeled *OaPAC*₁₋₁₀₂ which exists as a monomer and *OaPAC*₁₋₁₂₆ WT which forms a dimer due to downstream capping helix interactions. The results are shown below in Figure C2 and several conclusions can be drawn. First, the reviewer is correct that the presence of downstream domains indeed impacts the Trp conformation distribution. Comparing the WT of 1-102 and 1-126 in Figure C2a, we observed a drastic change in the W_{out} conformation, showing an increase from 15% to 81%. This phenomenon can be attributed to dimerization induced by the capping helix, as shown in the Figure C2b. In the dimer, Trp90 with the W_{out} configuration in one monomer interactions with Phe103 in the other monomer, which significantly increase the percentage of W_{out} conformations, whereas no such prominent hydrophobic interaction of Trp with residues on the capping helix is found in *SyPixD*. This suggests that Trp90 in *OaPAC* 1-126 is not suitable for probing the H-bond network in the active site. Second, In the Y6F mutant, the $W_{inNH_{in}}$ conformation also becomes predominant with a percentage of 88% even in the presence of the downstream domain, albeit with an altered chemical shift and several sub-peaks, which indicates that the capping helix may be flexible and introduces structural heterogeneity around the active site.

In summary, our results on the BLUF domains may not actually reflect the real Trp conformation in full-length proteins due to possible interaction of Trp with downstream domains (although the same ^{19}F NMR method could be used to obtain such information), yet our purpose is to use Trp conformation as a readout to reflect the difference in the H-bond network in the active site and to reveal the structural reason behind the diverse FMN* quenching, therefore we focus on the BLUF domain moieties of AppA and *OaPAC* in this study.

We have added one sentence in Paragraph 2, Page 21 in the Method section as follows:

“Full-length AppA₁₋₄₅₀ and *OaPAC*₁₋₃₆₆ contain additional downstream domains. In light of the potential influence of downstream domains on the conformational

distribution of Trp (Supplementary Fig. 15), which can render fluorinated Trp less suitable as a readout, we have constructed AppA₁₋₁₂₄ and OaPAC₁₋₁₀₂, with a focus on the BLUF domain.”

Figure C2. (a) ¹⁹F solution NMR results of 5-fluorotryptophan (5FW) labeled WT of OaPAC₁₋₁₀₂ and OaPAC₁₋₁₂₆ and corresponding Y6F mutants. (b) The W_{out} configuration of OaPAC₁₋₃₆₆ contacts Phe103 of the partner chain (PDB code: 4YUT).

4) Event though F has similar size as H, the electronegativity is different, thus the authors need to show that 5-FW and 7FW are not changing the behavior of the proteins and the fluorine is not controlling the conformation of the trp. The author should characterize 5-FW and 7-FW variants using TA to make sure the photocycle is not perturbed. They also can do the same experiment as in the supplemental table 3, KIE.

Author reply:

We thank Reviewer 1 for the important comments. We conducted additional transient absorption (TA) experiments on 5FW-labeled WT and 7FW-labeled WT in AppA, OaPAC and SyPixD, respectively. The results of the 760 nm kinetics are displayed in Figure C3. Both 5FW and 7FW exhibited kinetics that closely resembled each other. Further analyses involving various species, comprising both natural and non-natural variants, revealed that the trends remained consistent. This indicates that substituting natural tryptophan with non-natural fluorotryptophan did not introduce any discernible alterations to the proton transfer (PT) network. It is worth highlighting that in the case of AppA, the presence of both 5FW104 and 7FW104 resulted in a faster response. This

can be attributed to the higher proportion of $W_{in}NH_{in}$ in these variants, and the nearby Trp directly introduces an additional electron transfer (ET) channel. Conversely, in *OaPAC* and *SyPixD*, the kinetics of W90F and W91F closely mirrored their respective fluorinated W kinetics, in line with their shared predominance of proton relay configurations ($W_{in}NH_{out}$ and W_{out}).

We have added the above discussion in the manuscript in Paragraph 1, Page 6.

“Before delving further into the discussion, we conducted TA experiments to assess the impact of 5FW as well as 7-fluorotryptophan (7FW) (Supplementary Fig. 1). Remarkably, the kinetics at 760 nm revealed only minimal alterations with the same cross-species trend, affirming that the central photocycles remained unaltered upon fluorotryptophan incorporation.”

Figure C3. Overlapping of the kinetics at 760 nm of W_nF , 5FW labeled WT and 7FW labeled WT in AppA, *OaPAC* and *SyPixD*, respectively.

Sub-Section 2: Minor comment:

5) We use FAD and not FMN for BLUF protein.

Author reply:

We thank Reviewer 1 for the comments. Indeed, BLUF protein expression usually generates a heterogeneous distribution of flavin cofactors including FAD, riboflavin and FMN (Laan *et al.*, *Photochem. Photobiol. Sci.*, **2004**, 3, 1011) and a homogeneous incorporation requires hours of incubation of excess flavin molecule in interest. In our study, we opted for FMN over FAD due to concerns that the extra adenine moiety might induce a branching electron transfer pathway, potentially leading to flavin quenching as reported in the system of photolyases and cryptochromes (Zhong *et al.*, *Arch. Biochem. Biophys.*, **2017**, 632, 158), which would complicate our kinetic analysis on

the central photocycle.

We have added one sentence in Paragraph 1, Page 22 in the Method section as follows:

“After sonication, the centrifuged supernatant was incubated with FMN for at least 45 minutes. We opt for FMN over FAD in order to eliminate potential electron transfer pathway between lumiflavin and adenine within FAD.”

Reviewer 2:

We thank Reviewer 2 for stating “I think that this study determined, in the clearest way, the heterogenous positions/orientations of the important tryptophan residue in BLUF domain.” The responses to the reviewer’s comments are as follows.

1) *Section of "Ultrafast spectroscopy for YnF reveals a $W_{in}NH_{in}$ configuration" or "Solution NMR with site-specific 5FW probes for AppA, OaPAC and SyPixD"*

There is no description that explains that YnF mutant is not photoactivatable and/or FMN-Gln-Tyr is no longer used for the PCET

Author reply:

We appreciate the reviewer in suggesting providing a clear explanation regarding the YnF mutant’s photoactivatability and the change in the PCET pathway when FMN-Gln-Tyr is no longer used.

We have updated the text in Paragraph 2, Page 6 to include the following:

“Here we emphasize that the Y21F mutant is not photoactivatable and the central FMN-Gln-Tyr is no longer used for the PCET.”

2) *References I think that authors should cite the paper which first discovered the BLUF protein, such as "Masuda et al., Cell, 2002, 110, 613-23".*

Author reply:

We appreciate the reviewer’s suggestion, and we have included three earliest papers that first discovered the BLUF protein in the manuscript including the one mentioned by the reviewer, as well as Watanabe *et al.*, *Nature*, **2002**, 415, 1047 and Ono *et al.*, *Biochemistry*, **2004**, 43, 5304. These references are added in the manuscript as ref 9-11.

“Hence, structural heterogeneities are ubiquitously found in photoreceptors and

enzymes, such as in green fluorescence proteins (GFP)^{3,4}, phytochromes^{5,6}, microbial rhodopsins⁷, dihydrofolate reductases (DHFR)⁸, blue light using flavin (BLUF) domains⁹⁻¹¹, and liver alcohol dehydrogenases (LADH)⁸.”

3) Section *"Distinguishment and quantification for the W_{out} and $W_{in}NH_{out}$ configurations."*

I agree that the peak 1 of ^{19}F NMR corresponds to $W_{in}NH_{in}$ state, and the peak 3 corresponds to the W_{out} state, on the basis of the experimental data. However, the evidence to assign the peak 2 to $W_{in}NH_{out}$ state seems to be weak. I think that some evidence will be necessary to support the assignment of the peak 2.

Is peak 2 dependent on the D_2O percent?

Or, if the quantum chemical calculations using QM/MM method or some reasonable model of the active site qualitatively reproduce the ^{19}F NMR peak positions of $W_{in}NH_{in}$ and $W_{in}NH_{out}$, that would be sufficient for the assignment.

Author reply:

This comment made by Reviewer 2 is very insightful and we thank Reviewer 2 for carefully interpreting our results. The assignment of Peak 2 is indeed less straightforward than Peak 1 and 3, which was mainly deduced by the method of elimination and by correlating with the $W_{in}NH_{out}$ subpopulations in the X-ray crystal structures of SyPixD. However, we have performed the D_2O dependent experiment suggested by Reviewer 2 as shown in Figure C4 with 10% D_2O and 90% D_2O . It is obvious that both the solvent-exposed free 7FW peak and Peak 3 experience a slight yet noticeable shift from 10% D_2O to 90% D_2O , yet the positions of Peak 1 and Peak 2 are independent of the D_2O percentage. These results suggest that the Trp conformation corresponding to Peak 2 is also situated within the protein scaffold, similar to Peak 1, and can support our assignment of $W_{in}NH_{out}$.

We thank Reviewer 2 for suggesting quantum chemical calculations using QM/MM method. But unfortunately, at this time, we do not have the capability to conduct quantum chemical calculations. We hope to collaborate with computational chemists interested in this system in the future.

We have added a paragraph in the text on Page 15. “We conducted D₂O dependent experiment in SyPixD (Supplementary Fig. 12), the Peak 3 as well as free 7FW upshifted upon D₂O increase, yet the positions of Peak 1 and Peak 2 are independent of the D₂O percentage, which indicates both Peak 1 and 2 correspond to Trp conformations buried within the protein scaffold. At this point, we assign Peak 1 to W_{in}NH_{in}, Peak 2 to W_{in}NH_{out} and Peak 3 to W_{out}. Further quantum chemical calculations^{66,67} and solvent paramagnetic relaxation enhancement experiment⁶⁸ may provide additional support for these assignments.”

Figure C4. The ¹⁹F spectra for 7FW labeled SyPixD WT and free 7FW in D₂O/H₂O buffer containing 10% D₂O and 90% D₂O.

4) *It is not clear why 10% D₂O is used.*

Author reply:

We appreciate the reviewer’s question regarding using 10% D₂O in our experiments.

In NMR spectroscopy, modern spectrometers often employ deuterated solvents, such as D₂O, to stabilize the magnetic field strength. This stabilization is crucial for high-quality NMR measurements. Here is how it works: any fluctuations in the magnetic field, which can occur over time, are detected by the deuterium receiver. This receiver then adjusts the field strength (referred to as the “lock frequency”) to maintain a stable magnetic field.

In our experiments, we used 10% D₂O in the solvent to ensure the stability and accuracy of the NMR measurements by employing the deuterium lock. This practice is a standard procedure in NMR spectroscopy to obtain reliable and precise data

(Schwalbe *et al.*, *J. Vis. Exp.*, **2021**, 172, 1).

We have added the above information in Method section (Paragraph 2, Page 22) as follows.

“For the 700MHz NMR measurement, the protein sample was exchanged into 25 mM NaCl, 12.5 mM Na₂HPO₄/NaH₂PO₄ (pH 8.0) buffer using SDG-25 column chromatography and then lyophilized into powder overnight and re-solubilized in 10% D₂O (pD 8.0) as a locking agent⁷⁷, reaching a final concentration of 1.0~1.5 mM.”

5) *Trp_{in}* is used in the introduction, but *W_{in}* is used in other sections

Author reply:

We appreciate the reviewer’s comment. To maintain clarity and consistency throughout the manuscript, we have changed all “Trp_{in}” in the introduction into “W_{in}”.

Reviewer 3:

We thank Reviewer 3 for the highly positive comments “The results are convincing and conclusions are clear. I believe this manuscript is suitable for publishing on Nat. Comm”. The responses to the reviewer’s comments are as follows.

1) *The manuscript is focusing on the question about the multi-phasic relaxation of FMN*. The authors may need to introduce with a few sentences why it is important. Excitation of the FMN eventually results in the BLUF domain change to other conformations. Does the FMN relaxation relate to the photoisomerizing properties of BLUF domain?*

Author reply:

We appreciate the reviewer’s suggestion and agree that it is essential to provide context and explain the significance of the multi-phasic relaxation of FMN* in the BLUF domain. Indeed, the FMN relaxation relates to the photoisomerizing properties of BLUF domain. As proposed, blue-light excited FMN oxidizes a nearby Tyr, coupled with a proton relay via the intervening Gln in the forward PCET. Gln presumably isomerizes and flips before or upon a much elusive reverse PCET and lands in a so-called “light state”, wherein the hydrogen bond (H-bond) surrounding FMN is

rearranged, driving downstream conformational changes. These processes are at the heart of the BLUF domain's ability to sense and respond to blue light, making the understanding of FMN relaxation dynamics a critical component of unraveling the photoisomerization properties of BLUF domains.

In our revised manuscript, we include a brief introductory section that highlights the importance of this phenomenon in Paragraph 2, Page 3.

2) Related to question 1, the authors only gave the transient absorption spectra for the three proteins. How is the differences of the general photoswitching properties between AppA, SyPixD and OaPAC?

Author reply:

We thank the reviewer for the insightful comment. The BLUF domain is widely present in proteins of many microorganisms, and their downstream functions vary (for example, AppA mediates the DNA binding activity of PpsR through its SCHIC domain, while OaPAC regulates the catalytic adenylate cyclase functions through its AC domain). However, despite potential differences in downstream functions, the photoswitching functionality of the BLUF domain itself, is remarkably robust, in all known species. These suggest that the BLUF domain, as a photoswitch, may share an unknown unified photoactivation mechanism. However, as mentioned in the manuscript, researchers found that the ultrafast photochemistry of the BLUF domains in different species is quite diverse and controversial, including 1) the different FMN* quenching rates and 2) the various reaction intermediates captured in different species, making it difficult to converge to a conserved mechanism. Hence, our work aims to uncover the structural reasons (*i.e.*, the active-site H-bond network heterogeneities) behind these differences in photochemistry, which we believe will facilitate the ultimate discovery of the unified mechanism.

We have added some of the above explanations in Paragraph 2, Page 3.

3) Although fluoro-substitution is generally recognized as a minimal perturbation to protein structure, it affects the electron density distribution on the indole group of Trp.

Thus, the fluoro-Trp may affect the photochemical process compared with the real wild type protein. If I understand correctly, it seems like the authors used ^{19}F -Trp labeled protein for ^{19}F -NMR, but use the none labeled protein for transient spectral measurement. Thus, at least one transient spectral measurement needs to be done using the 5F-Trp labeled protein. Otherwise, the ^{19}F -NMR results might difficult to correlated with the transient spectra due to the unknown effect of fluoro-substitution.

Author reply:

We appreciate Reviewer 3 for raising this pertinent question. In response to this valid concern, we conducted transient absorption (TA) experiments with 5FW and 7FW in three different species. The results are presented in Figure C3, as this question was also raised by Reviewer 1. Notably, the kinetics at 760 nm revealed only minimal alterations with the same cross-species trend, affirming that the central photocycles remained unaltered upon fluorotryptophan incorporation.

4) The Y21F mutation abolishes the photo sensitivity of AppA (Biochemistry, 2003, 42(22):6726-34). However, the NMR spectrum of both WT and Y21F AppA showed only $W_{\text{in}}NH_{\text{in}}$ state (Peak 1). How to explain this? The authors may need to give the rationale in the manuscript. The authors may consider to measure the ^{19}F -NMR spectrum of AppA under low temperature to see is the observed peak is a coalesced peak of multiple peaks due to a fast exchange in NMR time scale.

Author reply:

We thank Reviewer 3 for the comments. About the first question regarding the NMR spectra of WT and Y21F AppA, it is important to note that in the early stages of research in the BLUF domain field, it was once believed that the photoactivation of the protein from the dark to the light state was accompanied by a change from W_{in} to W_{out} or W_{out} to W_{in} . For example, in one study, Bauer *et al.*, *Biochemistry*, **2006**, 45, 12687, it was suggested that W_{in} moves to W_{out} upon illumination. Contradictorily, in another work, Schlichting *et al.*, *J. Mol. Biol.*, **2006**, 362, 717, it was proposed that W_{out} flips to W_{in} upon photoactivation.

However, over time, researchers have discovered numerous exceptions to this notion.

For instance, in work by fluorescence (Kennis *et al.*, *Biophys. J.*, **2008**, 95, 312; Bauer *et al.*, *Biochemistry*, **2009**, 48, 9969), Raman (Unno *et al.*, *Biophys. J.*, **2010**, 98, 1949) and isotope-edited vibrational spectroscopic experiments (Mathes *et al.*, *J. Phys. Chem. Lett.*, **2015**, 6, 4749), it was indicated that the transition from dark to light does not always coincide with a transition between W_{in} and W_{out} .

Similarly, when key amino acids in active site of the BLUF domain are mutated to abolish the photoactivation function, it does not always lead to a W_{in} and W_{out} transition. The outcome depends on the nature of the mutated amino acid and how it affects the H-bond network in the active site. In the case of YnF, for instance, since Y cannot form a hydrogen bond with Q, Q can instead form a hydrogen bond with W, which leads to favoring the $W_{in}NH_{in}$ conformation in the YnF mutants across nearly all species.

Regarding Reviewer 3's second concern, we performed ^{19}F NMR experiments on 5-fluorotryptophan (5FW) labeled AppA WT at 4 gradient temperatures (i.e., 278K, 283K, 288K and 293K) with free 5FW as reference. The results are displayed below in Figure C5, Peak 1 remain a single peak at low temperature, only with a slight change in chemical shift and an increase in full width at half maximum (FWHM): Attributed to its uneven electron distribution and high nuclear charge, the ^{19}F nucleus exhibits a significant chemical shift anisotropy (CSA) effect (Case *et al.*, *Prog. Nucl. Magn. Reson. Spectrosc.*, **1998**, 32, 165). Rapid molecular motion under high temperature can effectively "smooth out" the CSA interactions, leading to narrower lines. In our case, ^{19}F in the protein molecule exhibits slower tumbling and rotation motions as the temperature decreases, thus Peak 1 get broader. However, we do not observe multiple peaks at the lower temperature, suggesting the observed peak at room temperature is not a coalesced peak of multiple peaks due to a fast exchange in NMR time scale.

We have added some of the above discussion on Page 6 and added the temperature-dependent experiment in the supporting information (Supplementary Fig. 2).

Figure C5. ^{19}F solution NMR results of 5-fluorotryptophan (5FW) labeled AppA WT under 278 K, 283 K, 288 K and 293 K with free 5FW as a reference.

5) In line 93, the use of 2HFOD and 2HFOB is very confusing. Using “(PDB code: 2HFO), chain D and chain B, respectively” might be better.

Author reply:

We thank the reviewer for the valuable suggestion. We have updated the caption for Fig. 1 on Page 5 as follows:

“ $W_{\text{in}}N_{\text{H}_{\text{out}}}$ (b) and W_{out} (c) configurations are adopted from the two subunits of SyPixD tetramer (PDB: 2HFO), chain D and chain B, respectively.”

6) The authors may need to give a note for YnF in the first time it appears.

Author reply:

We thank the reviewer for the suggestion. In response to this, we have made the following clarification on Page 7, Page 9 and Page 16:

“These akin chemical shifts for Peak 1, the congruous peak shift patterns from WT to YnF (*i.e.*, Y21F for AppA, Y6F for *OaPAC*, Y8F for SyPixD).”

“Hence, in the W_{out} configuration, FMN* presumably undergoes a nanosecond intersystem crossing (ISC) to form ^3FMN similarly to what happened for the YnF/WnF

(i.e., Y21FW104F for AppA, Y6FW90F for *OaPAC*, Y8FW91F for *SyPixD*) double mutant^{14,53}.”

“The 760 nm single wavelength kinetics for the three WnF (i.e., W104F for AppA, W90F for *OaPAC*, W91F for *SyPixD*) BLUF domains were analyzed using two approaches.”

7) *Please make sure the full name is given in place where the abbreviation is used for the first time. For example, TA appeared on Line 163, but the full name is at Line 447.*

Author reply:

We appreciate for the reviewer’s valuable feedback. We have carefully reviewed the entire manuscript and have made the necessary revisions to ensure that the full names are provided when abbreviations are used for the first time. We hope these changes enhance the readability and comprehensibility of the manuscript.

Below is a list of instances where we have added the full names:

- nuclear magnetic resonance spectroscopy (NMR), Paragraph 2, Page 4
- 5-fluorotryptophan (5FW), Paragraph 1, Page 5
- wild type (WT), Paragraph 1, Page 6
- transient absorption (TA), Paragraph 2, Page 4
- 7-fluorotryptophan (7FW), Paragraph 1, Page 6
- proton transfer (PT), Paragraph 1, Page 9
- two-dimension (2D), Paragraph 2, Page 9
- 4-fluorotryptophan (4FW), Paragraph 1, Page 15
- 6-fluorotryptophan (6FW), Paragraph 1, Page 15

8) *Temperature has significant influence on the thermodynamic properties, i.e., state populations. However, I didn’t see the temperature values anywhere in the manuscript.*

Author reply:

We appreciate the reviewer in pointing out this important concern regarding the absence of temperature values in the manuscript. To address this issue, we have added the experimental temperature in the appropriate sections of the manuscript where they

are essential, *i.e.*, captions for Figure 2 and Figure 5:

“Fig. 1: ^{19}F solution NMR results of 5FW labeled WT and YnF mutants of AppA, *OaPAC*, and SyPixD BLUF domains and 5FW alone in 10% D_2O at 298 K.”

“Fig. 5: ^{19}F solution NMR results of 7FW labeled WT samples of AppA, *OaPAC*, and SyPixD BLUF domains and 7FW alone in 10% D_2O at 298 K.”

The information of temperature is also added in the Method section: Paragraph 2, Page 22.

“Before loading into the NMR tube, the protein sample was centrifuged for 5 min at room temperature to remove any precipitates. Fluorine NMR spectra were recorded at 298 K using Bruker Avance Neo 700 MHz NMR Spectrometer (Bruker, Germany).”

9) *More details of ^{19}F -NMR spectral processing may need to be added. For example, how did you do the peak deconvolution?*

Author reply:

We thank Reviewer 3 for the suggestion. We used Lorentzian lineshape fitting to analyze the ^{19}F NMR spectra, minimizing the deviation of the residual sum of squares to achieve accurate peak deconvolution, which is shown in Figure C6: (as well as in Supporting Information Page 24, Supplementary Fig. 18).

The Lorentzian function used in the ^{19}F NMR deconvolution is represented as follows:

$$L(\omega, \omega_0, \gamma) = \frac{2}{\pi\gamma} \frac{1}{\left[1 + 4\left(\frac{\omega - \omega_0}{\gamma}\right)^2\right]}$$

Here, ω_0 is the location of the peak of the distribution, and γ serves as a scaling parameter specifying the full width at half maximum (FWHM) (Wüthrich *et al.*, *Science*, **2012**, 335, 1106).

The Lorentzian function also added in the header of Supplementary Table 1:

“Supplementary Table 1. Spectrum deconvolution as the sum of several Lorentzian peaks (Supplementary Fig. 19 for Fig. 2 and Supplementary Fig. 20 for Fig. 5) using

$L(\omega, \omega_0, \gamma) = \frac{2}{\pi\gamma} \frac{1}{\left[1 + 4\left(\frac{\omega - \omega_0}{\gamma}\right)^2\right]}$, ω_0 is the location of the peak of the distribution,

and γ serves as a scaling parameter specifying the FWHM. The estimated chemical shifts ω_0 and line widths γ are given in ppm. The areas under the peaks are provided as percentages (%). The chemical shift of free 5FW and 7FW are -124.8 and -134.2 ppm, respectively.”

Figure C6. Deconvolution of the 5FW ^{19}F NMR spectra (Fig. 2) into several Lorentzians, the estimated chemical shifts, line widths and the percentages of the areas under the peaks can be found in Supplementary Table 1.

10) Assignment of the multiple ^{19}F -NMR peaks is very challenging. I would suggest using solvent PRE to see the differential solvent exposure of peak 1, 2/2'; and 3/3';. This effect might be better observed by T1 relaxation measurement in the presence of caged gadolinium reagent such as Gd-DTPA-BMA. I would imagine that peak 3 and 3'; should have fastest solvent PRE effect.

Author reply:

We greatly appreciate your insightful comment regarding the assignment of the multiple ^{19}F -NMR peaks in our study. Solvent PRE (Paramagnetic Relaxation Enhancement) plays a crucial role in elucidating solvent interactions within biomolecules, with applications including the study of protein-binding interfaces (Prosser *et al.*, *J. Am. Chem. Soc.*, **2005**, 127, 5826), protein dynamics (Madl *et al.*, *J.*

Biomol. NMR, **2019**, 73, 305), and protein-protein interactions (Madl *et al.*, *Angew. Chem. Int. Ed.*, **2011**, 50, 3993).

We concur with your assessment that this technique has the potential to offer significant insights into our research. Specifically, the T1 relaxation time of the ^{19}F atom in the solvent-exposed W_{out} conformation will be reduced by adding soluble paramagnetic agents (*i.e.*, Gd-DTPA-BMA) in which their unpaired electrons can generate a magnetic field. At the same time, the PRE effect is less pronounced for the W_{in} conformations due to their long distance from the paramagnetic agents. Comparing different relaxation rates in a series of experiments with increasing concentrations of the paramagnetic agent, we could distinguish whether different peaks correspond to solvent exposure.

While we acknowledge the benefits of applying the solvent PRE method in this study, the T1 measurement is beyond the technical capability of our group. However, we are committed to advancing our research (currently we are also conducting ^{19}F NMR experiments by substituting the essential Tyr into fluoro-Tyr) and will explore opportunities for collaboration that may allow us to implement these techniques in the future.

We have added one sentence on Page 15:

“Further quantum chemical calculations^{66,67} and solvent paramagnetic relaxation enhancement experiment⁶⁸ may provide additional support for these assignments.”

11) *A supporting figure to show the sequence alignment of AppA₁₋₁₂₄, OaPAC₁₋₁₀₂ and SyPixD₁₋₁₅₀ would be very helpful and allow readers to see the conservative residues quickly.*

Author reply:

We thank Reviewer 3 for the suggestion. The sequence alignment figure is provided below and added into the SI (Paragraph 3, Page 5). The critical positions of the central Y, Q and the nearby W mentioned in the manuscript are labeled with black arrows.

Figure C7. The sequence alignment of AppA₁₋₁₂₄, OaPAC₁₋₁₀₂ and SyPixD₁₋₁₅₀.

REVIEWERS' COMMENTS

Reviewer #1 (Remarks to the Author):

The manuscript by Zhou et al investigates how the structural heterogeneity of the semi-conserved trp affects the photochemistry mechanism across BLUF proteins. They incorporated specific ^{19}F -Trp analogues in AppABLUF W64F, PixD and OaPACBLUF and using ^{19}F NMR combined with ultrafast spectroscopy, the author quantified the Trp configuration distribution in the 3 mentioned BLUF proteins. According to the authors, the conformational heterogeneity in the ground-state plays a crucial role in determining the observed differences in the initial light-activated step in BLUF domains. Their results show that AppABLUF compared to PixD and OaPACBLUF does not form radical(s) because the Trp is in WinNHin configuration, preventing a PT-coupled ET reaction from Tyr to FMN*. The authors concluded that the quenching rate is determined by the % of WinNHin. AppABLUF contains 100% WinNHin thus the longest quenching lifetime compared to OaPAC and PixD. OaPAC compared to PixD has slightly higher quenching rate due to higher proportion of WinNHin configuration.

The authors showed compelling experimental evidence that the conformational heterogeneity in the ground-state plays a crucial role in determining the difference in photochemistry. Even though the authors did an excellent job addressing my comments and providing additional information, the authors still cannot make the claim that they settled the long-term debate regarding the active-site heterogeneities in the dark-adapted state across different BLUF species. The OaPAC construct (1-102) used in their studies does not actually reflect the real Trp conformation in full-length proteins. We can clearly see that in the supplementary Figure 15 where the authors labeled OaPAC1-126 construct with 5FW and observed a drastic change in the Wout conformation, showing an increase from 15% to 81% and WinNHin confirmation decreased from 28% to 5%. According to these distribution of the Trp in OaPAC1-126, I would expect the quenching lifetime to be faster in OaPAC1-126 than PixD and not slightly slower. PixD compared to AppABLUF and OaPAC is a standalone BLUF protein which contains Trp close to its native form and OaPAC 1-126 is better construct for comparison because the trp environment resembles the full-length OaPAC.

Reviewer #2 (Remarks to the Author):

The authors adequately responded to my comments. I think that their study gives a clear answer to the challenging question about BLUF photoreceptors.

Reviewer #3 (Remarks to the Author):

The authors conducted necessary experiments and revised the manuscript accordingly. It addressed my concerns and I recommend this manuscript to publish on Nat. Comm.

The point-by-point responses to the reviewers' comments.

Reviewer 1:

We thank Reviewer 1 for stating that “The authors showed compelling experimental evidence that the conformational heterogeneity in the ground-state plays a crucial role in determining the difference in photochemistry.” We deeply value the expertise and thoughtful consideration of this complex research area. The response to the reviewers' comment is as follows.

- 1) *Even though the authors did an excellent job addressing my comments and providing additional information, the authors still cannot make the claim that they settled the long-term debate regarding the active-site heterogeneities in the dark-adapted state across different BLUF species. The OaPAC construct (1-102) used in their studies does not actually reflect the real Trp conformation in full-length proteins.*

Author reply:

We thank Reviewer 1 for this critical comment.

We have deleted the statement in Paragraph 2, Page 4 as follows:

~~“Together we settle the long-term debate regarding the active-site heterogeneities in the dark-adapted state across different BLUF species and uncover how structural heterogeneities affect the kinetics of the central photo-reactions.”~~